# Fibronectin-guided migration of carcinoma collectives

Sandeep Gopal[1,*,†], Laurence Veracini[1,*], Dominique Grall[1], Catherine Butori[2], Sébastien Schaub[1], Stéphane Audebert[3], Luc Camoin[3], Emilie Baudelet[3], Agata Radwanska[1], Stéphanie Beghelli-de la Forest Divonne[1,4], Shelia M. Violette[5], Paul H. Weinreb[5], Samah Rekima[1], Marius Ilie[2], Anne Sudaka[1,4], Paul Hofman[2] & Ellen Van Obberghen-Schilling[1,4]

Functional interplay between tumour cells and their neoplastic extracellular matrix plays a decisive role in malignant progression of carcinomas. Here we provide a comprehensive data set of the human HNSCC-associated fibroblast matrisome. Although much attention has been paid to the deposit of collagen, we identify oncofetal fibronectin (FN) as a major and obligate component of the matrix assembled by stromal fibroblasts from head and neck squamous cell carcinomas (HNSCC). FN overexpression in tumours from 435 patients corresponds to an independent unfavourable prognostic indicator. We show that migration of carcinoma collectives on fibrillar FN-rich matrices is achieved through αvβ6 and α9β1 engagement, rather than α5β1. Moreover, αvβ6-driven migration occurs independently of latent TGF-β activation and Smad-dependent signalling in tumour epithelial cells. These results provide insights into the adhesion-dependent events at the tumour–stroma interface that govern the collective mode of migration adopted by carcinoma cells to invade surrounding stroma in HNSCC.

[1] Université Côte d'Azur, CNRS, Inserm, Institut de Biologie Valrose (iBV), Parc Valrose, 06100 Nice, France. [2] Université Côte d'Azur, Laboratoire de Pathologie Clinique et Expérimentale, Biobank [BB-0033-00025] CHU Nice-Pasteur, 06001 Nice, France. [3] Aix Marseille Univ, CNRS, INSERM, Institut Paoli-Calmettes, CRCM, Marseille Protéomique, Marseille, France. [4] Centre Antoine Lacassagne, 06189 Nice, France. [5] Biogen Inc., Cambridge, Massachusetts 02142, USA. * These authors contributed equally to this work. † Present address: Department of Anatomy and Developmental Biology, Monash Biomedicine Discovery Institute, Monash University, Melbourne, Australia. Correspondence and requests for materials should be addressed to E.V.O.-S. (email: vanobber@unice.fr).

Head and neck cancer is the fifth most common malignancy reported worldwide with a 5-year survival rate of 50%, largely due to locoregional spread and recurrent disease following treatment failure[1]. Squamous cell carcinomas arising from stratified squamous epithelial tissues account for >90% of these malignancies[2]. In addition to the stepwise accumulation of genetic lesions within the target epithelium following carcinogen exposure, malignant progression of these tumours relies on functional interplay between tumour cells and their pro-tumoural tissue environment. The extracellular matrix (ECM) is a key component of the tumour microenvironment. Beyond providing support for cell adhesion/migration, it transmits chemical cues via signalling receptors of the integrin family and constitutes a platform for integrating the action of growth, chemotactic, angiogenic and immunomodulatory factors by regulating their distribution, activation and bioavailability[3]. Excessive synthesis and deposition of matrix proteins, a hallmark of the carcinoma-associated stroma, is primarily mediated by myofibroblasts, also referred to as carcinoma-associated fibroblasts (CAFs)[4].

Particular attention is being paid to the deposit of collagen by CAFs, in accordance with the emerging paradigm that the deposit and crosslinking of thick aligned fibres in the desmoplastic stroma of epithelial tumours is associated with aggressive tumour behaviours[5,6]. In normal tissues, fibronectin (FN) forms the provisional matrix that provides the framework for the assembly of fibrillar collagens as well as other ECM proteins[7,8]. We know from classical wound healing studies that matrix deposition follows a temporal sequence that begins with the assembly of FN, followed by cell invasion, deposition of type III then type I collagen before loss of FN[9]. In tumour tissue, often viewed as a non-healing wound, upregulated expression of FN is sustained. Early immunohistochemical analyses of head and neck squamous cell carcinomas (HNSCCs) showed FN overexpression at the invasive front and in the tumour stroma[10]. FN upregulated in tumours, referred to as cellular or 'oncofetal' variants (as opposed to plasma FN), harbours alternatively spliced exons encoding the highly conserved FN type III 'extra' domains A (EDA) and/or B (EDB). FN-EDA is a marker of the tumour vasculature[11] and a principal component of the metastatic microenvironment coined 'premetastatic niche' of a variety of tumours[12]. FN-EDB expression has been documented in several tumour types, including HNSCC[13,14] for which multivariate analysis showed a trend towards significant lower overall survival (OS)[15].

In the present study we sought to identify specific interactions between HNSCC cells and their ECM, and to address how these interactions affect cell migration. Tumour cell motile and invasive behaviours in three-dimensional environments have been extensively investigated *ex vivo* using gels composed of basement membrane proteins (Matrigel) or collagen I lattices from different sources with varying porosity and stiffness[16,17]. Although these studies provide valuable insights into molecular circuitries that regulate cell migration, de-cellularized ECM produced by mature cultures of fibroblasts, albeit complex, more faithfully re-capitulates the composition and architecture of the desmoplastic stroma of human tumours[6]. Therefore, we employed a binary *in vitro* system composed of tumour cells and a fibroblast-derived ECM that enabled us to: (i) characterize the motile behaviour of tumour cells, (ii) obtain a comprehensive ECM signature of HNSCC-associated fibroblasts and (iii) investigate relevant cell–ECM interactions and downstream signalling events.

We report that oncofoetal FN is a major component of the ECM assembled by HNSCC-associated fibroblasts and that its overexpression in the stroma of human tumours predicts adverse outcome. Remarkably, we identify $\alpha9\beta1$ integrin, and not $\alpha5\beta1$, as the $\beta1$-containing integrin involved in collective migration of

carcinoma cells on fibroblast-derived ECM. Finally, we show that $\alpha v\beta6$ integrin-stimulated migration occurs independently of direct TGF-$\beta$ activation/signalling in tumour cells.

## Results

**FN overexpression is associated with poor clinical outcome.** To examine the expression of FN in human HNSCC, a tissue microarray (TMA) was constructed with sample cores from 435 resected tumours (patient data, Table 1). FN immunostaining was positive in 136 tumours (31%) and weak or non-detectable in 299. FN overexpression, defined as moderate or strong staining, as depicted in Fig. 1a (scores 2–3), was significantly associated with the pathological tumor-node-metastasis (pTNM) stage and tumour grade but not with gender or tumour site (Table 1). Univariate survival analysis revealed that FN overexpression significantly correlated with lower disease-free survival (DFS) and OS in HNSCC patients with both early (high versus low FN; DFS, mean 37.6 versus 57.8 months, $P = 0.0001$; and OS, mean 40.5 versus 61.3 months; $P = 0.0001$) and late-stage disease (high versus low FN; DFS, mean 16.8 versus 31.6 months, $P = 0.0001$; and OS, mean 17.7 versus 34.5 months, $P = 0.0001$; Fig. 1b). In addition, a multivariate survival analysis using the Cox's proportional hazard model was conducted to examine the importance of FN expression in patient outcome when other prognostic factors were included. FN overexpression, the clinical stage, the pT status and pN status were significant independent prognostic factors for poor overall DFS and OS (Supplementary Fig. 1a).

FN was predominantly situated in the stromal compartment in these tumours where it co-localized with cellular FN isoforms harbouring the EDA domain (Fig. 1c). We confirmed that FN staining in TMA histospots from central areas of the tumour faithfully reflected FN expression in the remaining tumour tissue

**Table 1 | Clinical characteristics of head and neck cancer patients (n = 435) and correlation of cliniopathological variables with fibronectin status.**

|  |  | Fibronectin status |  |  |
| --- | --- | --- | --- | --- |
|  | *n* | − | + | *P*-values |
| *Patient cohort* |  |  |  |  |
| Age (years): 35–87 (median 68) | 435 | 299 | 136 |  |
| *Positive control* |  |  |  |  |
| Interpretable for fibronectin | 13 | 0 | 13 |  |
| No interpretable | 2 |  |  |  |
| *Gender* |  |  |  |  |
| Male | 360 | 244 | 116 | 0.345 |
| Female | 75 | 55 | 20 |  |
| *Site* |  |  |  |  |
| Oral cavity | 95 | 60 | 35 | 0.260 |
| Pharynx | 235 | 169 | 66 |  |
| Larynx | 105 | 70 | 35 |  |
| *pTNM stage* |  |  |  |  |
| I | 97 | 73 | 24 | 0.002 |
| II | 119 | 93 | 26 |  |
| III | 121 | 77 | 44 |  |
| IV | 98 | 56 | 42 |  |
| *Grade* |  |  |  |  |
| Well differentiated | 129 | 90 | 39 | 0.006 |
| Moderate differentiated | 208 | 156 | 52 |  |
| Poorly differentiated | 98 | 56 | 42 |  |

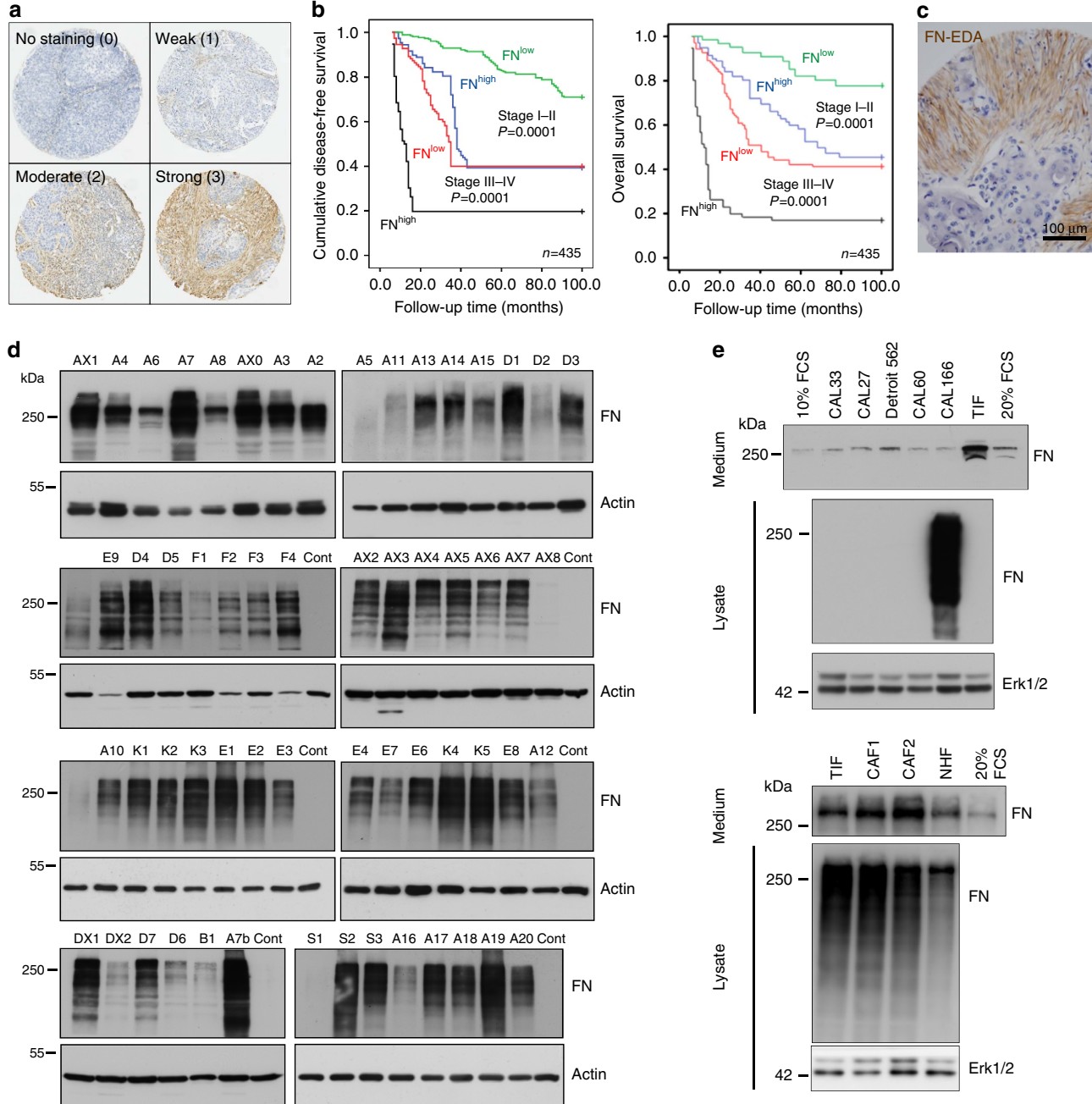

**Figure 1 | Fibronectin overexpression in human tumours predicts poor clinical outcome in HNSCC patients.** (**a**) Representative staining of total FN in histospots (600 μm) from HNSCC TMAs scored from 0 to 3. (**b**) Kaplan–Meier estimates of overall and DFS stratified by the dichotomized score of FN expression (low = score 0–1 versus high = score 2–3) in stage I–II (green and blue curves) and stage III–IV (red and black curves) tumours. *P*-values are from a log-rank test. (**c**) Representative staining of EDA-containing FN in a histospot. Scale bar, 100 μm. (**d**) Western blot analyses of FN expression in lysates of tissue samples from 60 human HNSCC (see Methods). Each blot includes six to eight tumour samples (patient numbers indicated above blots) and a control (Cont) lysate (40 μg, CAL33 cells). Actin staining provides a control of sample integrity and cellularity of tumour samples. (**e**) Western analysis of conditioned medium (top blots) and lysates (bottom blots) from HNSCC lines (CAL33, CAL27, Detroit 562, CAL60 and CAL166), TIF, CAFs and normal human fibroblasts (NF). Equivalent amounts of serum-containing culture medium (10% FCS for tumour cells and 20% FCS for TIFs) were deposited to determine levels of serum-borne FN. Erk1/2 expression in cell lysates was monitored as loading control.

by staining whole sections from a subset of tumours (*n* = 40; Supplementary Fig. 1b,c). Finally, western analyses were performed to assess the integrity of the molecule in tumour tissue. In lysates from a cohort of 60 human tumours (Fig. 1d), FN was detected at varying levels as a broad smear containing a discrete ladder of bands that corresponding, at least in part, to cleaved fragments. This pattern reflects the dynamic remodelling of the ECM in tumour tissue, consistent with an upregulation of FN-cleaving proteinases by both malignant and non-malignant cells[18]. A similar pattern was obtained for the cell/ECM-associated protein in lysates of early passage CAFs isolated from fresh tumours and hTERT-immortalized fibroblasts (telomerase-immortalized fibroblasts (TIFs)) from human skin (Fig. 1e). Although primary cultures of normal fibroblasts

express FN when cultured in presence of serum (a pathological milieu), the levels are considerably lower than those observed for TIFs and CAFs, and the presence of proteolysed fragments is greatly reduced (Fig. 1e). FN (secreted or assembled) was undetectable in a set of HNSCC lines (Fig. 1e), consistent with the observed restriction of FN expression to stromal cells in the majority of these tumours, and in line with previous reports[19].

**Abundant oncofetal FN in the HNSCC-derived fibroblast ECM.**
To study adhesive interactions between head and neck tumour cells and their ECM, we focused on fibroblasts, as they represent the major matrix-producing cells of the stroma, and exploited a pseudo-three-dimensional culture model in which carcinoma cells are plated on de-cellularized fibroblast matrices produced by CAFs or TIFs (Fig. 2a). Both CAFs and TIFs express α-smooth muscle actin, a marker of differentiated human tumour-associated myofibroblasts (Fig. 2b). Moreover, they both assemble a dense ECM with parallel fibre arrangement (Supplementary Fig. 2a), characteristic of the desmoplastic stroma of human HNSCC. To address mechanistic questions regarding cell–ECM interactions, it is crucial to obtain a detailed portrait of the matrix and matrix-associated proteins present in the de-cellularized ECM. Therefore, we took advantage of recent optimization of protocols for ECM preparation, together with the availability of a bioinformatic definition of ECM and ECM-associated proteins to do so[20]. Mass spectrometry-based proteomic analyses were performed on de-cellularized matrix from two independent CAF preparations, as described in Methods (Supplementary Fig. 2b,c). Results from theses analyses (Supplementary Data 1) were compared with the matrisome produced by TIFs, to validate the possible use of TIF-derived matrix for functional studies. As indicated in Fig. 2c, FN was among the major ECM glycoproteins produced by CAFs and TIFs. Other matrix glycoproteins include transforming growth factor-β-induced (TGFBI), Emilin-1 and Tenascin C (TNC). The CAF and TIF matrisomes contained a similar set of core matrix components and matrisome-associated proteins with comparable relative abundance, with the exception of type I collagen (α1 subunit), which was highly represented in the CAF1 matrisome (Fig. 2c and Supplementary Fig. 2). FN expressed and assembled by both CAFs and TIFs corresponds to cellular FN variants harbouring the alternatively spliced extra domains. Proteomic analyses allowed us to confirm the presence of FN-EDB−/EDA+ (isoform 1) and FN-EDB+/EDA+ (isoform 15) in the CAF1, CAF2 and TIF ECM proteomes by the identification of peptides specific to these two isoforms. Peptides specific to the FN-EDA− sequence were not detected. Indeed, staining of the FN matrix of CAFs and TIFs with an antibody specific for the EDA domain completely co-localized with staining of total FN, indicating that FN-EDA is incorporated in all of the FN fibres, whereas an FN-EDB-specific antibody recognized only a subset of fibres (Fig. 2b). Accordingly, FN transcripts expressed by TIFs contained relatively more of the EDA domain than the EDB domain (Fig. 2d).

Scanning electron microscopy and immunofluorescence analyses of the de-cellularized matrix revealed a dense fibrillar arrangement of matrisomal proteins (Fig. 2e and Supplementary Fig. 2a). Staining of TNC-containing fibres and collagen VI microfibrils in TIF-derived matrix partially overlapped with collagen I fibres, detected by second-harmonic generation imaging, whereas FN and collagen I were often localized in distinct fibrillar networks. This latter observation illustrates the dynamic partitioning of FN and collagen that occurs during ECM maturation (Fig. 2e).

**Persistent migration of cell collectives on TIF-derived ECM.**
We next analysed the migration of HNSCC cells on the fibroblast-derived fibrillar matrix. As TIFs produce a reproducible source of de-cellularized ECM and in light of the similar composition and organization of TIF- and CAF-derived matrices, we used cell-derived matrices assembled by TIFs for these analyses. Following adhesion to TIF-derived matrices, the HNSCC cells gather in clusters, with occasional strands and single cells aligned along the fibres (Fig. 3a,b). We previously showed that HNSCC cell clusters on fibroblast-derived ECM maintain expression and localization of junctional proteins. This can be seen by E-cadherin staining of CAL33 cells on a TIF-derived ECM (Fig. 3b and ref. 21).

Tracking of planar migration was assessed by live-cell imaging (Fig. 3c). Assuming that cell migration is a mixture of random and linear movements, directionality was determined by extracting the percent linear movement ratio from cell tracking results. Robustness of the algorithm was verified in a test model, depicted in Fig. 3d, in which tracks were simulated as a mixture of random walk and constant speed movement (as described in Methods). A significant increase in the speed and directionality of cell migration was observed on the fibrillar matrix, as compared with cells seeded on tissue culture-treated plastic (Fig. 3e). This indicates that cell clusters migrate faster in a more directional manner (oriented in a single direction). Hence, on the fibrillar lattice, cells move collectively with directional persistence and this occurs in the absence of a soluble chemotactic gradient (see Supplementary Movie 1). It is noteworthy that matrix metalloproteinase (MMP) expression and activity in conditioned medium is increased when cells are plated on cell-derived matrices, as compared with plastic (Supplementary Fig. 3), suggesting that MMPs could enable modification and/or release of motility-promoting factors from their pericellular environment. Indeed, punctate invadosome-like structures on tumour cells containing cortactin, and F-actin and tumour cell-derived extracellular vesicles were abundant at the cell–substrate interface on FN-rich fibrillar matrices, whereas cortactin staining was more diffuse and localized in lamellipodial-like actin-based protrusions in cells on adsorbed FN (Fig. 3b and Supplementary Fig. 4). No such accumulation of tumour cell-derived extracellular vesicles was observed on adsorbed FN.

**Cellular FN is essential for ECM assembly and cell migration.**
To determine the role of FN in motile and cohesive behaviour of cells on TIF-derived matrix, we generated FN-deficient fibroblasts expressing FN-targeting short hairpin RNA (shRNA) sequences[22]. The level of FN expression in these cells was substantially decreased, albeit not totally abolished (Fig. 4a). Whereas FN silencing had little or no effect on cell morphology or proliferation (Supplementary Fig. 5), ECM production by these cells was severely compromised. Only thin fibres and aggregates of residual FN could be detected after de-cellularization (Fig. 4b). Assembly of collagen-containing fibres was abrogated in FN-depleted conditions, confirming previous reports that FN assembly is essential for the deposition of other matrix fibrils, including type I collagen[7] (Fig. 4b). Although the results of this loss-of-function approach did not allow us to determine the role of FN in tumour cell migration, as no matrix was produced, our findings highlight the importance of FN expression by fibroblasts for orchestration of ECM assembly. To determine the role of collagen fibres, cell-derived matrices were prepared in the absence of ascorbic acid, a cofactor for hydroxylation of selected prolines and lysines involved in triple-helix assembly of fibrillar collagen. Under these conditions the TIF-derived matrix was markedly depleted in fibrillar collagen (Fig. 4c), whereas FN fibrillogenesis

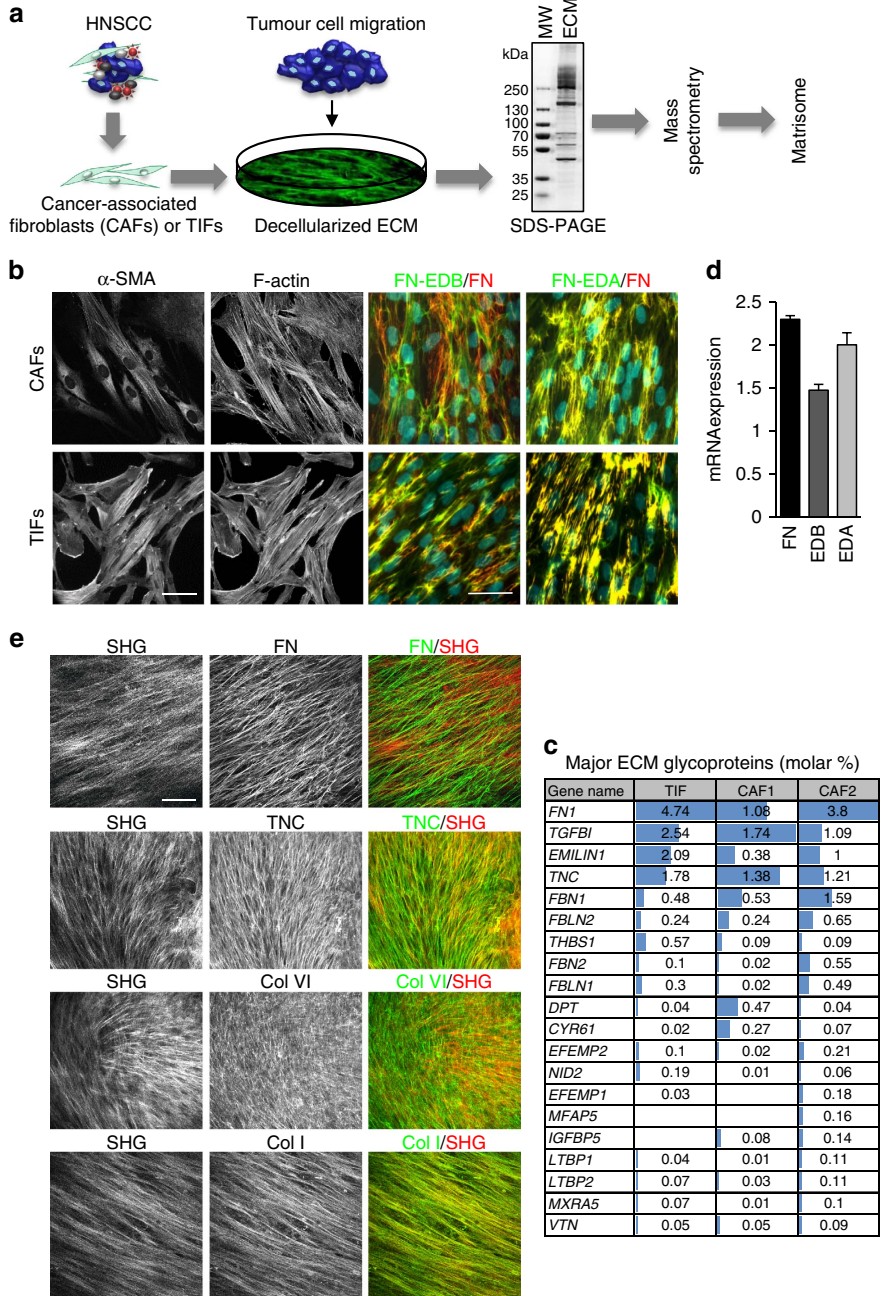

**Figure 2 | Composition and architecture of fibroblast-derived ECM.** (**a**) Scheme of the matrisome analysis workflow. (**b**) Left panels: representative immunofluoresence staining of α-smooth muscle actin (α-SMA) and F-actin in CAFs (top) and TIFs (bottom). Scale bar, 20 μm. Right panels: co-staining of total FN (green) and FN-EDB or FN-EDA/FN (red) in CAFs (top) and TIFs (bottom). Scale bar, 50 μm. Similar stainings were obtained for CAF1 and CAF2 preparations. (**c**) Top 20 ECM glycoproteins identified by mass spectrometry in cell-derived matrices produced by TIFs or by CAFs isolated from two different tumours. See Supplementary Data 1 for complete list of matrisome core proteins and matrisome-associated proteins. (**d**) Quantitative PCR analysis of transcripts encoding all FN isoforms (FN1), or isoforms harbouring the EDB and EDA, relative to GAPDH mRNA isolated from confluent TIFs. Results show mean ± s.d. from three independent experiments. (**e**) Comparison of second-harmonic generation (SHG) and immunofluorescence staining of TNC, FN, Col VI and Col I in TIF-derived matrix by two-photon laser microscopy. Representative images are from the same focal plane (n ≥ 3 fields per staining from at least three matrix preparations). Scale bar, 50 μm.

was only modestly affected. When tumour cells were plated on this collagen-poor ECM, we observed little or no differences in speed or directionality of migration, as compared with cells plated on matrices prepared in the presence of ascorbic acid (Fig. 4d,e). Similar results were obtained using another HNSCC line (Supplementary Fig. 6). Together, our findings suggest that interactions of tumour cells with the FN-rich fibres is sufficient to promote directional migration.

**αvβ6 and not α5β1 integrin controls collective migration.** We next asked which integrins were involved in collective migration of HNSCC cells on the TIF-derived ECM. First we determined the repertoire of integrins expressed by CAL33 cells and then we examined whether cell-surface expression of FN-binding integrins was increased in cells plated on FN-rich TIF-derived matrix, as compared with plastic. Messenger RNA expression profiles revealed high levels of transcripts encoding β1, β4, β5, α2, αv, α3

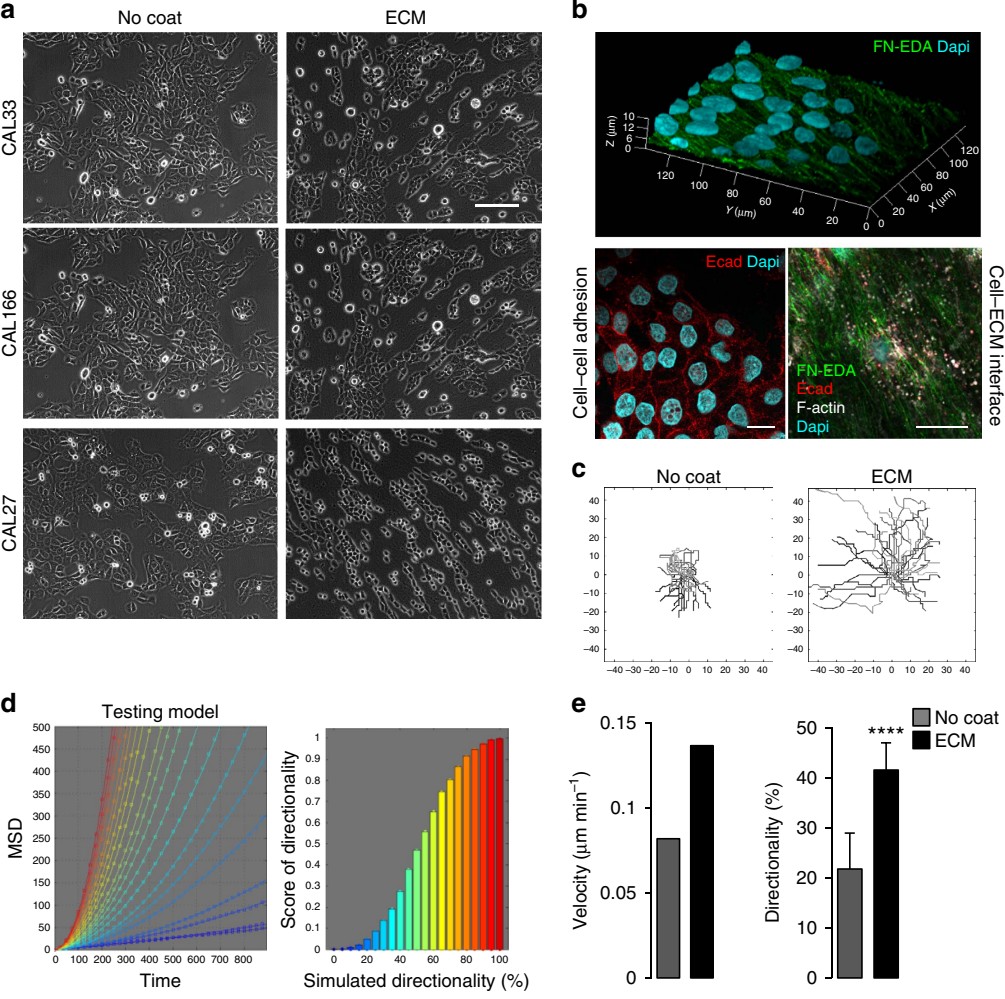

**Figure 3 | Directional migration of tumour cell cohorts on ECM fibres. (a)** Representative phase contrast images of HNSCC lines plated on plastic or TIF-derived ECM for 24 h. Scale bar, 150 μm. (**b**) Analysis of CAL33 cells on TIF-derived ECM by laser scanning confocal microscopy: (top) three-dimensional rendering of Dapi-stained nuclei and FN-EDA-stained ECM, (bottom) single projections of E-cadherin in intercellular junctions (left) and in tumour cell-derived vesicles sequestered in the ECM (right). Scale bar, 20 μm. (**c**) Migration of non-dividing CAL33 cells ($n \geq 100$) in cohorts was monitored by time-lapse videomicroscopy, 12 h after seeding on plastic or TIF-derived ECM. Images were acquired every 5 min for 24 h. Representative tracings (denoted by different grey levels) from origin of CAL33 cell cohort migration on plastic (No coat) or ECM are shown. (**d**) Tracks with different directionality ratios were simulated (mean square displacement (MSD) versus time), as described in Methods, and analysed in the same way as experimental data. (**e**) Histograms depict the velocity and directionality of cell movement on plastic (No coat) or ECM. For directionality, the confidence interval is measured as defined by MATLAB in the fit function (see 'Analysis of cell migration' in Methods). Statistically significant data are indicated by *$P < 0.05$, **$P < 0.01$, ***$P < 0.001$ or ****$P < 0.0001$. If no statistical difference, error bars are shown at 95%. Results from a representative experiment, of at least three, are shown.

and α6 integrin subunits, moderate levels of α5 and low or undetectable α1, β2 α4, β3, α7, α8, αL, αM transcripts in CAL33 cells (Supplementary Fig. 7a). mRNA transcripts encoding α9 and β6 integrin subunits (not present in the RT$^2$ Profiler PCR Array primer set) were also detected. Little or no change in integrin mRNA expression was observed between cells on plastic or ECM, with the exception of a 1.8-fold increase in β6 transcripts in cells plated on ECM (Supplementary Fig. 7b). However, fluorescence-activated cell sorting (FACS) analysis revealed a significant increase in cell surface expression of several FN-binding integrins, including the αv-based integrins αvβ6 and αvβ5, and a smaller increment in α5β1 when cells were plated on TIF-derived ECM (Fig. 5a). We were unable to assess surface expression of αvβ1, due to a lack of appropriately specific antibodies. This matrix-induced retention of integrins at the cell surface prompted us to examine the role of these FN-binding integrins in the regulation of collective migration.

Blocking α5β1, the prototype FN receptor and major FN-binding integrin in many cells, with a function-blocking antibody did not inhibit the migration of HNSCC cell cohorts (antibody control in Supplementary Fig. 8). Counterintuitively, it increased migration speed (Fig. 5b,c) and limited directional cell movement. This result suggests that α5β1 may reinforce adhesive interactions with matrix fibres and thereby impede collective cell movement, rather than enhance the motile behaviour of cells. No significant effect blocking α5β1 was observed on cell morphology or spreading of the HNSCC cells. In contrast, inhibition of αvβ6 integrin with a blocking antibody, severely reduced both speed and directionality of cell migration on TIF-derived matrix (Fig. 5b and Supplementary Movie 2). This inhibitory effect was mimicked using the αv integrin antagonist, S36578-2, known to target αvβ3, αvβ5 (ref. 23) and αvβ6 integrins (G.C. Tucker, personal communication). As shown in Fig. 5a, αvβ3 integrin is not expressed by these cells, consistent with its expression in the

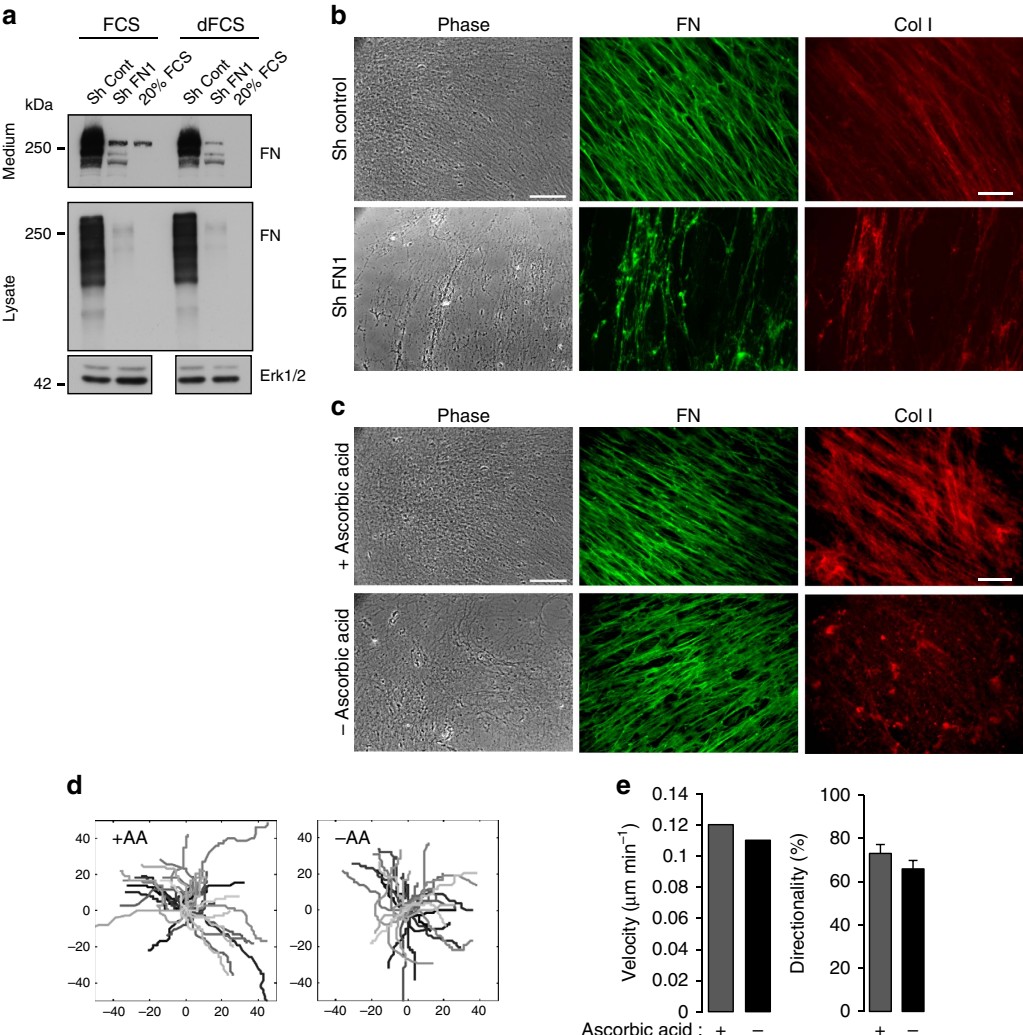

**Figure 4 | FN expression by TIFs is essential for matrix assembly and cell cohort migration.** (**a**) Western blot of FN in conditioned medium (top) and lysates (bottom) of TIF-derived cells stably expressing a control (Sh Cont) or FN-targeting shRNA (sh FN1) grown in presence of serum (FCS) or plasma FN-depleted serum (dFCS). Erk1/2 was monitored as loading control. (**b**) Left: phase-contrast images of de-cellularized ECM produced by control or FN-deficient TIFs cultured in medium supplemented with FN-depleted serum (scale bar, 150 μm); right: immunofluorescence staining of FN and collagen I fibrils in the ECM (scale bar, 15 μm). (**c**) Phase-contrast images and fluorescence staining, as in **b**, of TIF-derived matrix assembled by TIFs cultured in presence (+ AA) or absence (− AA) of ascorbic acid. (**d**) Representative trackings of cells (n≥100, denoted by different grey levels) within clusters seeded on ECM generated by TIFs cultured in presence (+ AA) or absence (− AA) of ascorbic acid. (**e**) Histograms depicting the speed and directionality of movement from a representative experiment of three are shown. Statistical methods for analysis of cell migration data are described in Methods. If no statistical difference, error bars are shown at 95%. Results from a representative experiment, of at least three, are shown.

endothelial rather than epithelial cells of these tumours[19]. Moreover, the use of αvβ5 blocking antibody had no impact on collective migration speed or directionality (Fig. 5d,e), demonstrating that in these cells the inhibitory effect of S36578-2 antagonist is likely to be due to the targeting of αvβ6 and possibly αvβ1. From these results we conclude that αvβ6 is necessary, and that α5β1 is dispensable for collective migration of HNSCC cohorts on TIF-derived matrix.

**αvβ6 regulates migration independently of TGF-β signalling.** The FN receptor αvβ6 was the first integrin to be identified as an activator of TGF-β[24]. Indeed, increased αvβ6 levels on the surface of CAL33 cells plated on TIF-derived ECM, as compared with plastic, correlated with a significant increase in the levels of total TGF-β1 (latent + active) released into the culture medium (Fig. 6a). This TGF-β1 corresponds to the active form, at least

in part, as we observed augmented phospho-Smad2 and -Smad3 in cell lysates (Fig. 6b) and induction of TGF-β-responsive genes (Fig. 6c). These results suggest that the adhesion of cells to the matrix can trigger both release of the cytokine from the ECM and its activation. TIF-derived matrix-induced Smad phosphorylation was dependent on αvβ6, as blocking the integrin with a functional-blocking antibody, or with the S36578-2 αv integrin antagonist, effectively blocked Smad2 phosphorylation (Fig. 6d) and this inhibition could be reversed by the addition of exogenous, active TGF-β1. In contrast αvβ5- and β1-blocking antibodies had no effect on Smad2 phosphorylation (Fig. 6d), demonstrating that activation of TGF-β signalling was specific to αvβ6. We next investigated whether αvβ6-induced TGF-β signalling was involved in collective migration of HNSCC cohorts. Although addition of active TGF-β1 could bypass the αvβ6 blockade of Smad signalling, it was unable to override the inhibitory effect of αvβ6-blocking antibodies on collective cell

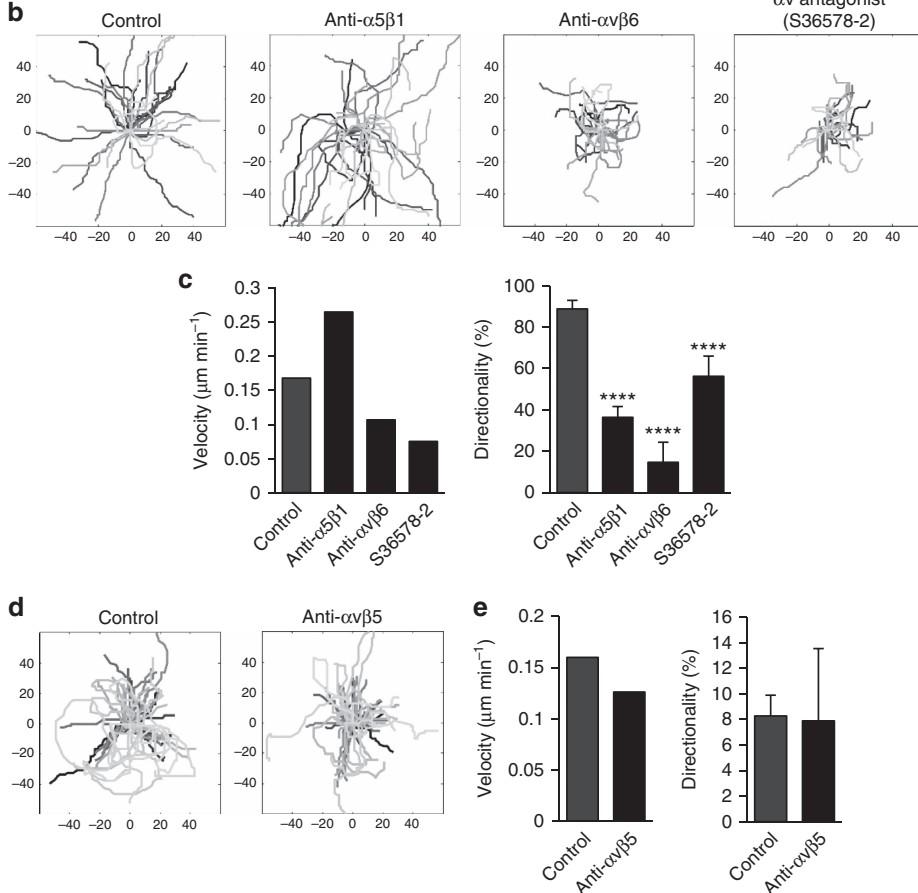

**Figure 5 | αvβ6 Integrin regulates collective cell migration on TIF-derived ECM.** (**a**) Surface staining (mean fluorescence intensity) of the indicated integrins was determined by flow cytometry. (**b–e**) Cells were allowed to adhere to ECM for 12 h before addition of the indicated blocking antibody. (**b,d**) Tracking of cells (denoted by different grey levels) within clusters seeded on TIF-derived ECM was performed for 6 h following addition, or not (Control), of blocking antibodies: anti-α5β1 (10 µg ml$^{-1}$, clone JBS5) anti-αvβ6 (45 µg ml$^{-1}$, Stromedix 6.3G9) anti-αvβ5 (20 µg ml$^{-1}$, clone P1F6), 5 µg ml$^{-1}$ of the S36578-2 αv integrin antagonist. (**c,e**) Histograms depict the velocity and directionality of movement. Statistical methods for analysis of cell migration data are described in Methods. Results from a representative of at least three independent experiments are shown.

migration (Fig. 6e,f). Moreover, addition of TGF-β1 had no effect on the speed or directionality of cell migration on TIF-derived matrix and pharmacological inhibition of Smad signalling with the TGF-β receptor I kinase did not interfere with ECM-stimulated collective migration (Fig. 6d–f). It is noteworthy that all of the HNSCC cell lines we examined were insensitive to growth inhibitory and EMT-inducing effects of TGF-β. As shown in Fig. 6, adding active TGF-β1 to the culture medium did not affect cell morphology (Fig. 6g) or inhibit proliferation, evidenced by histone H3 phosphorylation (Fig. 6h). Similarly, seeding CAL33 cells on TIF-derived matrix, which increases active TGFβ1, did not inhibit histone H3 phosphorylation (Fig. 6i). Taken together, these results support

a model in which αvβ6 regulates collective migration in HNSCC cells independently of TGF-β/Smad pathway activation.

**α9 is the key β1 integrin partner in HNSCC.** α9β1 is a β1-based integrin that binds to the EDA domain of FN in an RGD tripeptide-independent manner[25]. Based on the fact that the FN incorporated in CAF-/TIF-derived matrices contain the EDA domain, and the presence of EDA-containing FN in the stroma of head and neck tumour tissue, we hypothesized that α9β1 may be involved in the directional migration response observed on the TIF-derived matrix. Similar to the other FN-binding integrins, plating CAL33 cells on ECM induced an increased surface

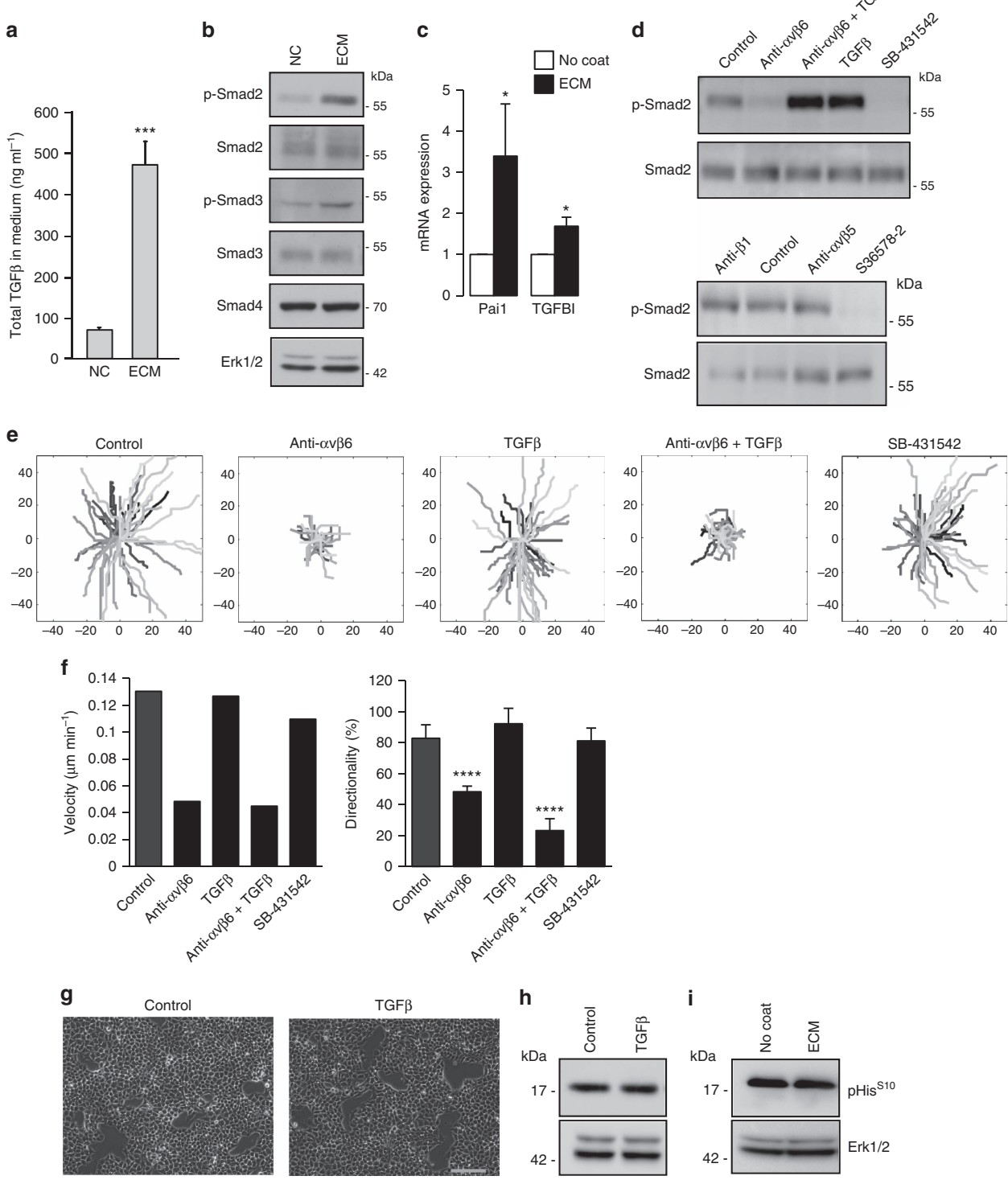

**Figure 6 | αvβ6 Integrin-dependent collective cell migration is independent of TGF-β activation. (a)** Total TGF-β1 secreted by CAL33 cells plated on non-coated plastic dishes (NC) or TIF-derived matrix (ECM) was measured by ELISA (mean ± s.d. from three independent experiments). **(b)** Western blotting of Smad2 (Ser465/467) and Smad3 (Ser423/425) phosphorylation and **(c)** qPCR analysis (mean ± s.d. from three independent experiments) of Pai1 and TGFBI mRNA expression in cells seeded for 36 h on ECM, versus plastic (No coat) dishes (*P < 0.05). **(d)** Western blot analysis of cell lysates from cells plated on ECM for 12 h and treated for 24 h with no addition (Control) or blocking antibodies to αvβ6 (45 µg ml$^{-1}$), αvβ5 (20 µg ml$^{-1}$), β1 (10 µg ml$^{-1}$) or S 36578-2 (5 µM), TGF-β1 (5 ng ml$^{-1}$) or TGFβRI inhibitor SB-431542 (10 µM). **(e)** Representative tracings of cells in clusters migrating on ECM (denoted by different grey levels) in the absence (Control) or presence of the indicated blocking antibody added 12 h after seeding. **(f)** Histograms depict the corresponding velocity and directionality of movement. See Methods for statistical analysis of migration data. Results from a representative experiment, of at least three, are shown. **(g)** Representative phase contrast images of CAL33 cells on plastic in the absence (Control) or presence (48 h) of TGF-β1 (5 ng ml$^{-1}$). Scale bar, 150 µm. **(h)** Western blotting of histone H3 phosphorylation on serine10 in lysates from non-treated (Control) or TGF-β1-treated CAL33 cells. **(i)** Western blot analysis of histone H3 phosphorylation on serine ten in lysates from CAL33 cells plated on ECM for 12 h.

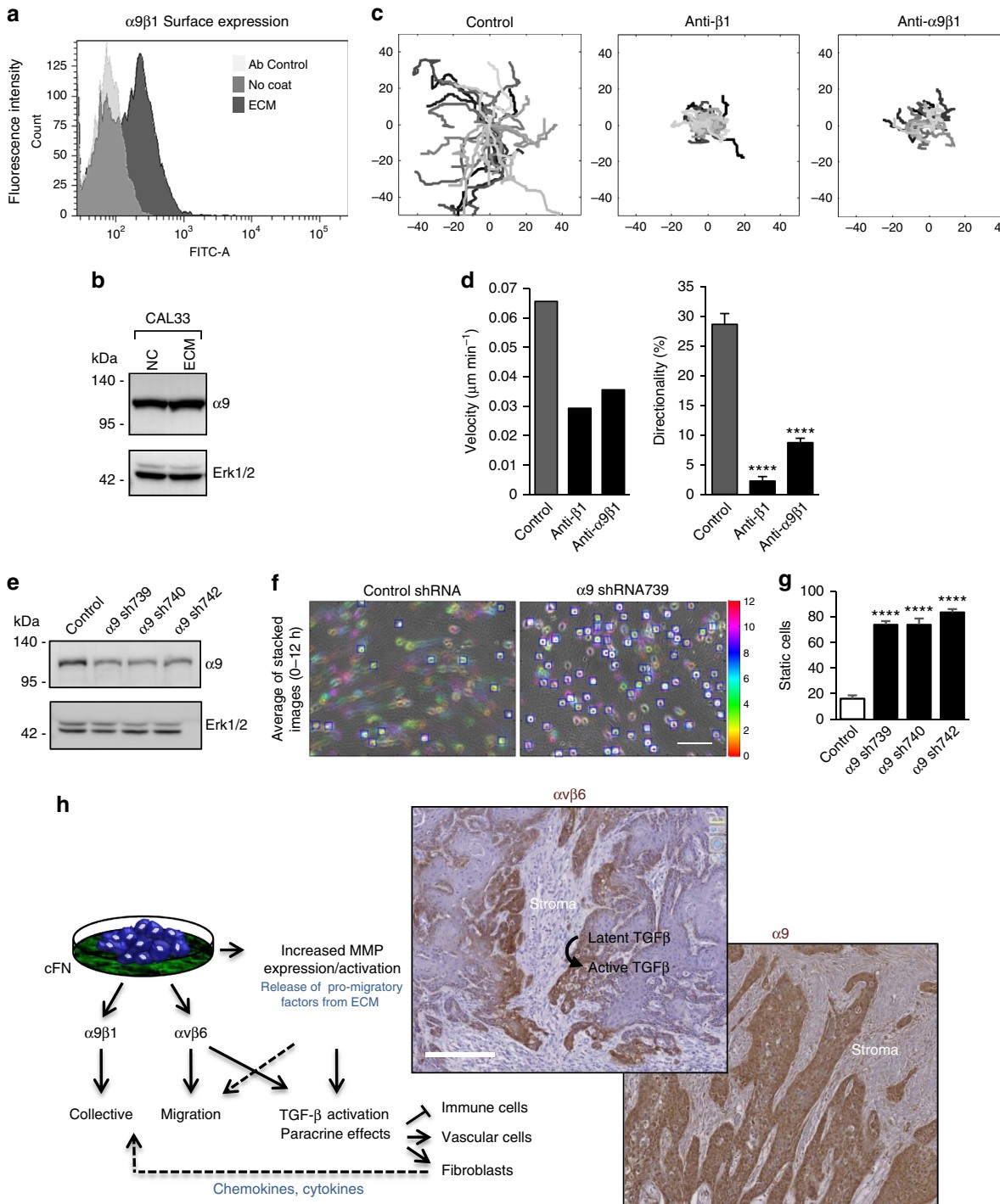

**Figure 7 | The β1-based integrin α9β1 regulates motility of HNSCC cohorts on ECM. (a)** Mean fluorescence intensity of α9β1 integrin expression on the surface of CAL33 cells was determined by flow cytometry (anti-α9β1 Clone Y9A2) in cells plated for 12 h on non-coated plastic (No coat) or TIF-derived ECM, as indicated; isotype control is shown by a light grey histogram. Data are representative of FACS analysis results from three independent experiments. **(b)** Western blotting of α9 integrin subunit expression in lysates of cells plated for 12 h on non-coated plastic (NC) or ECM. **(c)** Representative tracings from origin (denoted by different grey levels) and **(d)** histograms showing the velocity and directionality of CAL33 cell migration ($n \geq 100$ cells) on TIF-derived ECM in the absence (Control) or presence of function blocking antibodies to α9β1 (10 μg ml$^{-1}$, clone Y9A2) or β1 (10 μg ml$^{-1}$, clone P5D2) integrins. Results from three experiments are shown; see Methods for description of statistical analysis. **(e)** Western blotting of lysates from CAL33 cells expressing control (Control shRNA) or three different α9-targeting shRNA sequences. Erk1/2 was monitored as loading control. **(f)** Motion analysis of control or the indicated α9-depleted CAL33 cells on TIF-derived ECM. Average of stacked phase contrast images from videomicroscopy 0–12 h after plating. Static components are in greyscale (static cells are framed in blue); cell motion is coloured chronologically. Scale bar, 150 μm. **(g)** Quantification of static cells (mean ± s.d. from three fields) expressing three different shRNA sequences. Results from one of three independent experiments are shown. Statistically significant data are indicated by *$P<0.05$, **$P<0.01$, ***$P<0.001$ or ****$P<0.0001$. **(h)** Left: schematic summary of our results and (right) representative staining of αvβ6 and α9 in human HNSCC samples.

expression of this integrin (Fig. 7a), with no change in total α9 protein levels (Fig. 7b). Interfering with α9β1 binding to the matrix had a marked inhibitory effect on the speed and directionality of cell migration (Fig. 7c,d and Supplementary Movie 3), and the inhibition was nearly as extensive as the inhibition observed with β1 function-blocking antibody (Fig. 7c). This effect was specific to α9β1, as decreasing expression of the α9 integrin subunit through α9-targeting shRNA sequences (Fig. 7e) had a pronounced inhibitory effect on cell spreading and motile behaviour on the fibrillar ECM, as determined by motion analyses shown in Fig. 7f,g. These results are consistent with the notion that α9 is the β1 integrin partner involved in regulating HNSCC cell migration.

## Discussion

The first key message of our present work is that sustained expression of oncofetal FN in the stromal compartment of human HNSCC is strongly associated with decreased survival of patients. Our in vitro results underscore the essential role for autocrine FN in ECM assembly by stromal fibroblasts and in guiding directional migration of coordinated tumour cell groups, which is reminiscent of collective migration through stromal tissue in vivo (schematized in Fig. 7h). Further, they reveal a TGF-β-independent function for αvβ6 in promoting tumour cell migration along the FN-rich ECM, in addition to its role in activation of latent TGF-β that fuels the stromal TGF-β signalling programme at the tumour–stroma interface (Fig. 7h). Finally, they identify the α9 integrin subunit, expressed in HNSCC lines and human tumours (Fig. 7h)[26], as a key β1 integrin partner for this effect.

Our study provides the first comprehensive data set of the human HNSCC-associated fibroblast matrisome. In addition to identifying FN among the most abundant matrix components, our proteomic analyses offer an inventory of proteins that could engage in functional interactions in the HNSCC microenvironment. FN has been shown to interact directly with the major core matrisome proteins present in the CAF ECM, including TGFBI, TNC, thrombospondin1, collagens I/VI, fibrins and fibrillins, and to promote their incorporation in the ECM network[7,8,27–30]. TGFBI, a TGF-β-inducible protein associated to aggressiveness in several tumoural settings, is highly abundant in the HNSCC-associated fibroblast matrisome. Consistent with this observation, overexpression of the protein was observed in the stroma of human HNSCC (Supplementary Figs 9–11), in particular at the tumour–stroma interface in the vicinity of αvβ6-mediated TGFβ activation. In developing tissues, TGFBI has been described as a bifunctional linker between collagen VI microfibrils and cell surface receptors[31]. Similar to TGFBI type, VI collagen (second most prevalent collagen in the TIF/CAF matrisome) fulfils a bridging function between ECM components, including FN[32], as seen in our confocal imaging analyses. It is tempting to speculate that, beyond their structural role, these bridging molecules may have major implications in regulating the biomechanical and signalling properties of the HNSCC ECM. In contrast to TGFBI, periostin, a TGFBI paralogue and key premetastatic niche component[33], was detected at very low levels and in only one of the CAF matrisomes. Immunostaining of periostin in whole sections of HNSCC revealed variable expression in the stromal compartment of some tumours, which could explain the absence or low expression in our fibroblast matrisomes (Supplementary Figs 10 and 11). Together, these analyses of matrix composition and organization advance our understanding of the stromal ECM of HNSCC and in addition build a solid investigative framework for subsequent studies.

Directional migration of HNSCC cell collectives on the fibroblast-derived ECM is reminiscent of the migration of cell cohorts, which takes place during embryogenesis along pre-patterned chemoattractant gradients[34,35]. This raises the question of how it occurs. The ECM binds soluble growth factors and is often proposed to act as a source of tethered ligands that may assist in shaping external chemoattractant gradients. Indeed, FN alone is known to interact with a wide array of growth factors and chemokines[36]. Among the rather limited set of stromal-derived secreted factors in our detergent-extracted matrisomes, IGF2 and Wnt5A (a non-canonical Wnt ligand) were present in the matrix of both tumour-derived fibroblast preparations. Their ability to enhance directional motility of HNSCC cells and potential relevance for tumour dissemination in vivo merits further investigation. Plunging deeper into the stromal matrisome of human HNSCC using improved fractionation and proteomic-based approaches should reveal additional factors and generate new ideas about ECM-mediated guidance in collective cell migration and how to block it.

Importantly, the mode of carcinoma cell migration has been shown to be dependent on the TGF-β competence of the cells[37]. In a breast tumour model, cells that are competent for TGF-β signalling exhibit strand or single cell migration, whereas cells that have lost responsiveness to TGF-β (TGFβRII knockout tumour cells) migrate primarily in cell clusters. Consistent with this scenario, we observed collective migration of the examined HNSCC lines that are refractory to both the growth suppressive and mesenchymal reprogramming effects of TGF-β. It should be noted that in these same lines Smad signalling appeared intact, as treatment with TGF-β1 (or exposure to the matrix-bound form) stimulated TGFβRII-dependent phosphorylation of Smad2/3 and induced expression of select TGF-β target genes. Thus, Smad phosphorylation does not necessarily reflect TGF-β responsiveness with respect to growth and EMT induction. Rather, multiple parameters must be taken into account for determining TGF-β competence of carcinoma cells. It is well established that attenuation of the TGF-β signalling axis in HNSCC, as defined by mutation or downregulation of Smad4, occurs in up to 50% of human tumours (see ref. 38). Accordingly, a large proportion of HNSCC retain a cohesive epithelial phenotype at invading tumour fronts. Collectively, these observations underscore the complexity of TGF-β regulation and signalling, and illustrate the challenges in assessing responsiveness of epithelial tumour cells to the cytokine for therapeutic considerations.

CAFs represent a key source of TGF-β[39,40]. Not surprisingly, the most abundant components of the HNSCC CAF matrisome correspond to proteins that are either upregulated by TGF-β (for example, FN, TGFBI and TNC) or proteins involved in TGF-β storage and activation by cellular receptors (including FN, fibrillins and latent TGF-β-binding proteins). Of note, latent TGF-β activation in culture requires the presence of an FN matrix[41]. Integrin αvβ6 is a well-established activator of matrix-bound TGF-β[24]. Whereas the integrin is not detected on normal keratinocytes, its expression is strongly upregulated in head and neck carcinomas at the FN-rich tumour–stroma interface[42–44] (Fig. 7h). Increased levels in HNSCC lines have been associated with enhanced tumour cell migration, invasion and regulation of MMP expression (see ref. 45 and references therein). Our results that αvβ6-dependent migration of HNSCC cells on the cell-derived matrix is independent of TGF-β signalling demonstrate that alternative pathways mediate αvβ6-stimulated movement of cell cohorts on the matrix. Identifying these pathways could improve αvβ6-targeting therapeutic strategies if both proinvasive and TGF-β-activating functions of the integrin could be inhibited.

FN-binding integrins expressed by epithelial tumour cells include RGD-dependent integrins (α5β1 and αv-class integrins) that bind to the 'cell binding domain' of FN and α9β1 integrin that recognizes a distinct sequence in the alternatively spliced EDA domain. α5β1 is generally considered to be the β1-based integrin that drives invasive carcinoma migration in FN-rich matrices[46,47]. Indeed, the ability of this integrin to sense substrate rigidity and generate traction forces for migration in FN environments has been extensively studied (typically on plasma FN) in fibroblasts and in various tumour cells[47–51]. Viewed in this context, our findings that blocking α9β1, rather than α5β1, efficiently phenocopy the effect of pan-β1-blocking antibodies on cohort migration are unexpected and hence thought provoking. These results extend our understanding of how cell–matrix interactions control migration of adhesive cell cohorts and they open a new avenue of investigation. Indeed, α9β1 is one of the most recently identified and least characterized integrins[52]. It has been shown to enhance migration of certain cell types on various substrates, yet the underlying signalling events are not well understood. Reported mechanisms include the involvement of an inward rectifier potassium channel, Src-dependent nitric oxide synthase activity and vesicle trafficking[52]. However, little is known about the ability of α9β1 integrin to regulate cytoskeletal organizing pathways and support mechanotransduction. Decrypting the details of these pathways and their regulation by α9β1 should provide an important key to the understanding of the migratory behaviour of carcinoma collectives. In addition to the EDA domain of FN, other ligands present in the CAF matrisome, including TNC, EMILIN1 and thrombospondin-1, have been reported to bind α9β1 (ref. 52). It is conceivable that interactions with these ligands may also contribute to the motility-promoting effects of the integrin that we observe on fibrillar ECM.

Our findings support a role for α9 in human HNSCC. To date, clinicopathological associations between α9 and HNSCC are not established. However, our analysis of the limited set of publicly available data in GEO DataSets revealed that ITGA9 mRNA expression may be linked to tumour grade, pT and to HPV status. Future studies on larger patient cohorts will be needed to strengthen and extend these results.

To conclude, we have advanced the understanding of the tumour-stroma dialogue through characterisation of the stromally-generated matrisome and identification of two FN-binding integrins as key determinants for tumour cell motility in HNSCC. However, important unmet challenges remain. This progress in knowledge must now to be leveraged to exploit its potential diagnostic, prognostic and therapeutic utility.

## Methods

**Cell culture and ECM preparation.** The human head and neck cancer cell lines CAL33, CAL27, CAL166 and CAL60 were established in the Antoine Lacassagne Cancer Centre[53] and Detroit 562 cells were from American Type Culture Collection (Rockville, MD, USA). HNSCC lines were cultured in DMEM medium (Invitrogen, Cergy Pontoise, France) containing 10% (v/v) FCS. Normal human skin fibroblasts immortalized with the telomerase reverse transcriptase (*hTERT*) gene were provided by J. Norman (Beatson Institute, Glasgow, UK) and cultured in DMEM supplemented with 20% (v/v) FCS and 20 mM Hepes. HNSCC-derived fibroblasts were isolated as described[54]. Briefly, minced tumours were placed on scratches in a cell culture dish and maintained in DMEM containing 10% FCS (v/v), 100 units per ml penicillin, 100 μg ml$^{-1}$ streptomycin and 1 μg ml$^{-1}$ fungizone until the first passage. CAFs began to migrate from the tumour fragments after 2 weeks. At 70% confluence, tumour fragments were removed and the CAFs were transferred to new culture flasks. Only low (<3) passage fibroblasts were used for analysis to minimize cellular drift and contamination by dormant tumour cells.

All cells were routinely tested, with negative results, for mycoplasmal contamination by PCR (Mycoplasma Plus, Stratagene, La Jolla, CA). Cell-derived matrices produced by CAFs and TIFs were prepared as described[55]. FN-depleted FCS was obtained using gelatine sepharose-4B (GE Healthcare, Uppsala, Sweden) columns.

**Reagents.** DNase I was purchased from Roche (Penzberg, Germany). TGF-β1 was from R&D Systems (Abingdon, UK). SB-431542, the selective inhibitor of TGF-β receptor I (ALK5), and ascorbic acid were from Sigma-Aldrich (St Louis, MI, USA). The non-peptidic αv integrin antagonist (S36578-2) was kindly provided by G. Tucker and J. Hickman (Institut de Recherches Servier, Croissy sur Seine, France).

**Antibodies.** The following antibodies were used: polyclonal anti-Smad2, anti-phospho-Smad2 (Ser465/467), anti-Smad3 and anti-phospho-Smad3 (Ser423/425) from Cell Signalling Technology (Beverly, MA); monoclonal anti-FN and anti-E-cadherin from BD Biosciences (Le Pont de Claix, France); monoclonal anti-TNC (clone BC24), α-smooth muscle actin (clone 1A4), anti-β actin (clone AC-15) and polyclonal anti-periostin from Sigma-Aldrich; monoclonal anti-collagen I, blocking monoclonal α9β1 antibody (cloneY9A2), rabbit anti-pHis (Ser10) and anti-TGFBI from Abcam (Cambridge, MA); anti-FN-EDA (clone IST-9) from Sirius Biotech (Genoa, Italy); anti-ERK1 (clone C-16) from Santa Cruz Biotechnology (Santa Cruz, CA); polyclonal anti-FN, monoclonal anti-αvβ3 integrin (clone LM609), monoclonal anti-α5β1 integrin (MAB1999), blocking monoclonal anti-α5β1 integrin (clone JBS5), monoclonal anti-αvβ5 integrin (clone P1F6), monoclonal anti-αvβ6 integrin (clone E7P6), monoclonal anti-collagen VI, anti-cortactin p80/85 and monoclonal anti-α9β1 integrin were from Millipore (Billerica, MA); anti-β1 integrin (clone lia1/2) from GenWay Biotech (San Diego, CA); blocking monoclonal anti-β1 (clone P5D2) from R&D systems; polyclonal anti-α9 integrin from Thermo Scientific (Rockford, IL); and mouse monoclonal anti-CD31 from DAKO Corp. (Carpinteria, CA). Anti-αvβ6 integrin monoclonal antibodies for function blocking (clone 6.3G9) or immunostaining (clone 6.2A1) were prepared as described[44]. References of commercial primary antibodies and dilutions used in experiments are indicated in Supplementary Table 1.

Secondary antibodies coupled to horseradish peroxidase were from Jackson ImmunoResearch Laboratories, Inc. (West Grove, PA): anti-mouse IgG, 1:20,000, 115-035-062 and anti-rabbit, 1:40,000, 115-035-045. Alexa Fluor-congugated secondary antibodies were purchased from Invitrogen: Alexa Fluor 488 goat anti-rabbit, 1:500, A-11034; Alexa Fluor 546 goat anti-rabbit, 1:500, A-11035; Alexa Fluor 488 goat anti-mouse, 1:400, A-11029; and Alexa Fluor 546 goat anti-mouse, 1:400, A-11030. Vectastain ABC Kit (Vector Laboratories, Burlingame, CA) and Alexa Fluor 647 Phalloidin (1:100, A22287) were from Molecular Probes (Thermo Scientific).

**Lentiviral vector construction and transduction.** The pLB2CPGm lentiviral vector harbouring FN- or luciferase-targeting (control) shRNA sequences have been described[22]. pLKO.1-puro-based MISSION lentiviral vectors harbouring α9 integrin-targeting sequences (or pLKO.1 control vector) were purchased from Sigma-Aldrich. Lentivirus production and cell transduction were performed as described[22]. Targeting sequences are as follows:

FN-targeting shRNA sequences:
(FN-sh1) 5′-TTTCTGTTTGATCTGGACCTGC-3′
(FN-sh2) 5′-TATGCTTTCCTATTGATCCCAA-3′
(Cont-sh) 5′-AATTAGTCGCTGAAGTCCGCC-3′
Integrin α9-targeting shRNA sequences:
(739) 5′-CGAAGGTACAAAGAAATTATC-3′
(740) 5′-GCTGGAAGAACATCTACTAT-3′
(742) 5′-TGTGAAGAACATCTCCCTA-3′

**Western blot analysis.** Cells were lysed with Laemmli buffer and proteins were analysed by western blotting according to standard procedures. Western analysis of tumour tissue has been described[56]. Uncropped scans of representative blots are shown in Supplementary Figs 12–14.

**ELISA assay.** TGF-β1 secreted in the conditioned medium of cells cultured for 48 h on plastic or ECM was quantified using the ELISA Quantikine kit from R&D Systems according to the manufacturer's instructions.

**Fluorescence-activated cytometric analysis.** Cell surface integrin expression was determined by flow cytometry as described[57], using a FACSAria fixed-alignment benchtop high-speed cell sorter.

**Immunofluorescence and microscopy.** Immunostaining of cells or ECM was performed as described[21]. Fluorescence was observed through ×40/1.3 or ×63/1.4 oil objectives on a Zeiss inverted epi-microscope (Axiovert 200 M) equipped with an ANDOR NEO sCMOS camera. Second-harmonic generation and confocal imaging were performed on a multi-photon microscope (Zeiss NLO780) confocal system with a ×63/1.4 objective.

**Quantitative real-time PCR.** Reverse transcription was performed on TRIzol reagent-extracted RNAs with the High Capacity complementary DNA Reverse Transcription Kit (Applied Biosystems, Foster City, USA). FN1, EDB and EDA mRNA expression was quantified on a StepOnePlus real time PCR System using the TaqMan Fast Advanced Master Mix (Applied Biosystems) according to the

manufacturer's instructions. Expression of mRNA corresponding to PAI1 and TGFBI genes was determined by real-time PCR using the Fast SYBR Green Master Mix (Applied Biosystems). Fold stimulation was calculated using the ΔCT or ΔΔCT method, as indicated. TaqMan human gene-specific primers are as follows: GAPDH (Hs02758991_g1), FN1 (Hs01565277_m1), FN1-EDA (AJPAC8E; PN4441114) FN1-EDB (AJQJBEM; PN4441114), ITGB6 (Hs00982345_m1) and ITGA9 (Hs00979865_m1). SYBR Green primer sequences were as follows: PAI1, sense 5′-GACCGCAACGTGGTTTTCTC-3′ anti-sense 5′-CCTTGTACAG ATGCCGGAGG-3′; TGFBI, sense 5′-CTCCAGCCAACAGACCTCAG-3′ anti-sense 5′-ATAGACAGGGGCTAGTCGCA-3′.

**Patient cohorts.** For TMAs, 435 surgically removed tumours embedded in paraffin blocks were retrieved from the archives of the Pathology Department of the University Hospital of Nice. The cases, received between 1991 and 2007, included HNSCC exclusively from patients without metastases from whom consent was obtained, and clinical data with a follow-up of at least 8 years were available. Frozen HNSCC samples used for analysis of FN expression by western blotting were obtained from consenting patients included in the CARISSA multicentre blinded institutional review board-approved phase II trial of postoperative irradiation with cisplatin ± gefitinib (GORTEC 2004-02—NCT00169221). Clinicopathological data have been reported[56].

**TMA construction and immunohistochemistry and statistics.** TMA construction and analysis of tissue samples were performed in accordance with Institutional Guidelines. Haematoxylin and eosin-stained sections of primary tumours were reviewed, and areas of tumour and normal tissue were marked on the slides. Areas of necrosis and keratin pools were avoided. Representative carcinoma areas were selected for building TMAs and arrays were designed as previously described[58]. Briefly, three tissue cores (600 μm in diameter) corresponding to three representative central areas of the tumour were selected. The tissue cores were arrayed into a recipient paraffin block using a fine steel needle and an automatic tissue microarrayer (Beecher Instruments, Sun Prairie, WI, USA, and Alphelys, Paris, France). TMAs of primary carcinomas contained non-tumoural tissue adjacent to benign oral tumours from patients (six tissue cores from biopsies performed on these patients), which served as control and for regulating mark spacing between core centres; cores were spaced at intervals of 1 mm in the x and y axes. A 4 μm haematoxylin and eosin-stained section was reviewed to confirm the presence of tumour tissue and immunohistochemical methods were performed on serial 2 μm deparaffinized TMA sections. For FN detection, intrinsic peroxidase was blocked by incubating sections with 3% hydrogen peroxide for 6 min. Sections were blocked in 4% goat serum for 20 min, then incubated for 12 min with mouse monoclonal anti-FN (BD Biosciences) or anti-FN-EDA antibody (IST-9, Sirius Biotech). After rinsing with PBS, sections were incubated with biotinylated secondary antibody for 20 min, rinsed with PBS and incubated with StreptABComplex/HRP (DAKO Corp.) for 20 min. Sections were then washed with distilled water, incubated with developing solution (3-amino-9-ethylcarbazole; DAKO), counterstained with haematoxylin and mounted with aqueous mounting medium. After staining, slides were analysed by three pathologists (CB, MI and PH) using a multihead microscope. Stained slides were scanned on the Hamamatsu NanoZoomer 2.0-HT Digital slide scanner (40X mode) and images acquired using the NDP.view2 software (Hamamatsu Photonics K.K.,Hamamatsu City, Japan). A disc was considered unsuitable for analysis if it was completely absent, it contained no tumour tissue (sampling error), or it contained too few tumour cells (<10%) for analysis (uninformative). For each patient, the mean score of a minimum of two core biopsies was calculated. Whole-tissue sections from tumour blocks of a subset of 40 cases were stained and compared by visual inspection with the corresponding TMA cores using the above mentioned scoring. The FN expression levels observed in the TMA cores faithfully reflected the FN labelling in whole-tissue sections from corresponding tumour blocks (Cohen's κ coefficient = 0.94). Comparison of FN status with the clinical and pathological variables including gender, site, TNM stage and histological grade was made using $\chi^2$ analysis. Calculations and analyses were performed with SPSS 16.0 for Windows (SPSS Inc., Chicago, IL, USA) and where appropriate, were two tailed.

**Mass spectrometry analysis.** For proteomic analysis of de-cellularized matrices, gelatine coating of the surface was omitted from the matrix preparation protocol. We verified that this modification did not alter organization of the ECM or the motile behaviour of cells seeded onto it (Supplementary Fig. 2b,c). ECM prepared from TIFs or CAFs was frozen at −80 °C until further processing. Protein extracts were solubilized in urea, reduced and alkylated, and proteins digested with first PNGaseF (New England BioLabs, Ipswich, MA), endoproteinase Lys-C (Promega, Madison, WI) and high-sequencing-grade trypsin, as described by Naba et al.[20]. Samples (injected in triplicate) were reconstituted in 0.1% trifluoroacetic acid (TFA) 4% acetonitrile and analysed by liquid chromatography-tandem mass spectrometry (MS/MS) in an LTQ-Orbitrap-Velos (Thermo Electron, Bremen, Germany) online with a nanoLC Ultimate 3,000 chromatography system (Dionex, Sunnyvale, CA). Peptides were separated on a Dionex Acclaim PepMap RSLC18 column. First peptides were concentrated and purified on a pre-column from Dionex (C18 PepMap100, 2 cm × 100 μm I.D, 100 Å pore size, 5 μm particle size) in solution A (0.1% formic

acid–2% acetonitrile). In the second step, peptides were separated on a reverse phase column from Dionex (C18 PepMap100, 15 cm × 75 μm I.D, 100 Å pore size, 2 μm particle size) at 300 nl min⁻¹ flow rate. After column equilibration by 4% of solution B (20% water–80% acetonitrile–0.1% formic acid), peptides were eluted from the analytical column by a two steps linear gradient (4–20% acetonitrile/H₂0; 0.1 % formic acid for 90 min and 20–45–45% acetonitrile/H₂0; 0.1 % formic acid for 30 min. For peptide ionisation in the nanospray source, spray voltage was set at 1.4 kV and the capillary temperature at 275 °C. Instrument method for the Orbitrap Velos was set up in data dependant mode to switch consistently between MS and MS/MS. MS spectra were acquired with the Orbitrap in the range of m/z 400–1,700 at a FWHM resolution of 30,000 measured at 400 m/z. For internal mass calibration the 445.120025 ion was used as lock mass. The ten abundant precursor ions were selected and collision-induced dissociation fragmentation was performed in the ion trap on the ten most intense precursor ions measured to have maximum sensitivity and the maximum amount of MS/MS data. The signal threshold for an MS/MS event was set to 500 counts. Charge-state screening was enabled to exclude precursors with 0 and 1 charge states. Dynamic exclusion was enabled with a repeat count of 1, exclusion list size 500 and exclusion duration of 30 s.

**Protein identification and estimation of abundance.** The acquired raw LC Orbitrap MS data were processed using the MASCOT search engine (version 2.4.1)[59]. Spectra were searched against a SwissProt Human database (version 2015.11; 20 194 entries). The following parameters were used for searches: (i) precursor tolerance 6 p.p.m.; (ii) MS/MS tolerance 0.8 Da; (iii) trypsin using 2 missed cleavages; (iv) fixed modification of cysteines by carbamidomethylation ( + 57.02146); (v) variable modifications oxidation of methionine, proline and lysine ( + 15.99491), amino-terminal acetylation ( + 42.0106) and asparagine deamidation ( + 0.984016). The false discovery rate were set to 1% for identity and homology threshold, and determined by searching in parallel a decoy database. For protein grouping, all proteins that could not be distinguished based on their identified peptides were assembled into a single entry according to the Mascot rules.

The exponential modified protein abundance index was calculated automatically by Mascot to enable estimation of protein abundance[60].

**Analysis of cell migration.** Phase contrast and video microscopy were performed using a × 10/0.3 numerical aperture air objective. Image acquisition was performed using the MetaMorph Imaging System (Universal Imaging Corp., Downingtown, PA). Migration of cells was monitored by manual tracking of non-dividing cells (>100) in groups and analysed using the 'QuanTrack' MATLAB script (or standalone), developed in house and available on request.

Cell migration results from a combination of random (or Brownian) movement and persistent movement in one direction. If a cell migrates more randomly, directionality decreases and vice versa. To determine the relative contribution of these two components (directionality ratio $\rho_{Dir}$), we used the experimental mean square displacement curves ($C_{MSD}$). Thus, we fitted the $C_{MSD}$ with a linear combination of Brownian and constant speed movements. Given the fact that the Brownian movement is linear with time and the constant speed movement is quadratic with time, we defined the linear movement ratio ($\rho_{Dir}$) of each cell as $C_{MSD}(t, \rho_{Dir}, D) = 4(1 − \rho_{Dir})Dt + \rho_{Dir}v^2t^2$, where $v$ is the mean instantaneous speed. $\rho_{Dir}$ was fixed to vary from 0 (that is, Brownian movement) to 1 (directional movement) and is used as a directionality metric.

For directionality, the confidence interval is measured as defined by MATLAB in the fit function. Asterisks and error bars are defined for confidence bounds at 95% (**), 99% (***) and 99.9% (****). If no statistical difference, error bars are shown at 95%.

A testing model was used to confirm that our metric is robust and consistent. Thus, we simulated tracks as a mixture of Random walk and constant speed movement. More precisely, the movement of the cell was built as:

$$\mathbf{P_i}(t + \Delta t) = \mathbf{P_i}(t) + V_m * \Delta t((1 − \widetilde{\rho_{Dir}})\mathbf{M_{brown}}(t) + \widetilde{\rho_{Dir}}\mathbf{M_{dir}})$$

With: $\mathbf{M_{brown}}(t) = R_1(t)\begin{bmatrix}\cos(2\pi \times R_2(t)) \\ \sin(2\pi \times R_2(t))\end{bmatrix}$ and $\mathbf{M_{dir}} = \begin{bmatrix}\cos(\theta_i) \\ \sin(\theta_i)\end{bmatrix}$

$R_{1,2}(t)$ are randomly chosen at each iteration from 0 to 1 and $\theta_i$ a constant per track. By varying the directionality ratio in our simulations, we could link $\rho_{Dir}$ obtained from the analysis to $\widetilde{\rho_{Dir}}$ defined as a parameter of the simulation. The sigmoidal pattern of curve was expected.

The value obtained from the 'QuanTrack' MATLAB script for velocity measurements represents the speed distribution of all cells monitored in a given analysis. This distribution is Poissonian-like; therefore, by definition the s.d. is equal to the mean value and error bars are irrelevant.

Quantification of static cells was performed using stacked images acquired every 5 min (0–12 h following adhesion) by timelapse microscopy. We extracted a temporal variance map $Var(x, y) = \frac{1}{N_t}\sum_{t=1}^{N_t}(I(x, y, t) − \langle I(x, y, t)\rangle_{x,y})^2$ from which we extracted isolated donut-shape patterns (corresponding to cell size). For visualization, the displacement sequence is colour coded.

**Statistical analyses.** Student's two-tailed non-paired t-tests were used to determine the statistical significance for pair-wise comparison in the in vitro analyses shown in Figs 6a,c and 7g, and Supplementary Figs S2c, S7b and S8. For results

presented in Figs 3e, 4e, 5c,e, 6f and 7d, and Supplementary Fig. S6c, see 'Analysis of cell migration'. For both types of analyses, statistically significant data are indicated by $*P < 0.05$, $**P < 0.01$, $***P < 0.001$ or $****P < 0.0001$.

**Data availability.** The mass spectrometry proteomics data, including search result, have been deposited on the ProteomeXchange Consortium site (www.proteomexchange.org)[61] via the PRIDE partner repository with the dataset identifier PXD003457. The authors declare that all other data supporting the findings of this study are available within the paper and its Supplementary Information files or available from the corresponding author upon request.

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

## Acknowledgements

We thank the consenting patients and the head and neck surgeons of the Nice CHU-Pasteur and the IUFC for providing tumour samples. We are grateful to Gertraud Orend, Dean Sheppard and team members for valuable discussions. Katia Havet is kindly acknowledged for assistance with TMA construction and analysis, Christelle Bonnetaud, Emmanuel Chamorey and Renaud Schiappa for statistical analyses, Agnes Loubet for FACS analyses, Jim Norman for providing TIFs and Patricia Rousselle for normal human fibroblasts. We thank the iBV imaging (PRISM) and histopathology facilities for valuable assistance with cell imaging and tissue staining. Proteomic analyses were performed at the Marseille Proteomics facility (http://map.univmed.fr/). P.H., M.I., A.S. and E.V.O.-S. are affiliated to the FHU OncoAge (http://www.oncoage.org/), and P.H. and M.I. to the Hospital Biobank (BB-0033-00025), Pasteur Hospital, CHU Nice. Financial support was provided by the French National Cancer Institute, the Fondation ARC, the Ligue National Contre le Cancer (PAIR-VADS11-023), the Cancéropôle PACA, the foundation 'Cancer, Aidez la recherche!' and the LABEX SIGNALIFE program (ANR-11-LABX-0028-01).

## Author contributions

E.V.O.-S. conceived the project, analysed the data, established collaborations and allocated funding for the work. The manuscript was written by E.V.O.-S. and edited by all authors. S.G., L.V., D.G., S.B.-d.l.F.D., A.R. and S.R. performed all experiments involving molecular, cellular and biochemical analyses, performed immunohistochemistry on tissue sections and contributed to the experimental design and data analysis. S.A., L.C. and E.B. performed the mass spectrometry and analysis of data sets. S.S. developed image analysis software and contributed to the analysis of live imaging experiments. P.H., C.B. and M.I. generated the HNSCC TMA and provided pathological expertise. A.S. provided tumour samples and pathological expertise. S.M.V. and P.H.W. provided reagents for detection and functional analysis of αvβ6 integrin.

## Additional information

**Competing financial interests:** The authors declare no competing financial interests.

