## [Peer Review File · Nature Communications]

Reviewers' comments:

Reviewer #1 (Remarks to the Author): Expert in HNSCC and clinical analysis

This is an interesting manuscript that assesses contributions of micro-environment in the directional movement of head, neck squamous cell carcinomas. They identify oncofetal fibronectin as a key component of the matrix and associate its overexpression with poor outcome in patients with head neck squamous cell carcinomas. The findings are novel and of interest to a broad audience. There are several questions about the data and methodology that need to be addressed. A key question is why transform fibroblasts were used, rather than cancer associated fibroblasts for all of the experiments in the manuscript. Details about the figures and controls are inadequate. The statistical approach is not well described either in the main manuscript or supplementary methods for several of the experiments. While the conclusions at the authors drive are congruent with the data presented, there is insufficient controls to make firm statements about precise mechanisms being investigated in this work. While this manuscript is potentially interesting, several issues need to be addressed.

Major issues include:

1. Figure 1 details results from immunostaining for fibronectin in a cohort of head neck cancer patients on a tissue microarray. The results show an association between fibronectin expression and TNM stage, tumor grade, and survival. A key issue with these findings is the fact that these experiments were performed on tissue microarrays. The construction of TMA's typically focuses on identifying areas within a tumor that are densely populated by cancer cells. It is unclear whether the results from the TMA reflect the expression of fibronectin in the remainder of the tumor. Although the supplementary data states that whole sections were compared for 40 cases, results from this comparison are not reported in this manuscript. In addition, while the supplementary data section includes details about the statistical analyses performed in these experiments, only disease-free survival is provided in the manuscript. Overall survival and local recurrence or not given. Moreover, while it is indicated that both univariate and multivariate analyses were performed, results of multivariate analysis are not provided anywhere in the manuscript.
2. In addition, the authors suggest that fibronectin is primarily expressed by stromal fibroblasts. Based on Western blot drive expression patterns in primary tumors, cancer cell lines, and demoralized human fibroblasts. While this may be the case, I'm not sure they can make this conclusion based on the data presented.
3. In figure 2, the authors present data showing proteins deposited into matrix by transform fibroblasts and cancer associated fibroblasts. It would be of interest to know whether cancer associated fibroblasts also have patterns of fibronectin expression, similar to transform fibroblasts (as part of Western blot in figure 1E). If possible, it will be important to include a control of non-transformed fibroblasts to assure that the patterns of expression are specific to the cancer microenvironment.
4. Is it possible for the authors to assess whether fibronectin transcripts also contain more EDA domains, primary tumor samples.
5. In figure 3A, the difference between organization of CAL 133 on extracellular matrix does not look significantly different from those with no coat. A better image or multiple images should be provided to show this difference.
6. In figure 4, the requirement for fibronectin expression is assessed by knocking down fibronectin in transform fibroblasts and assessing effect on matrix formation. The authors state transform fibroblasts in which fibronectin is knockdown using shRNA do not produce a matrix. It would be important to know whether knocking down fibronectin in transform fibroblasts affects cellular activities essential to normal function and viability of the cells. If the cells are "sick" they may not produce a matrix solely because they lack fibronectin.
7. It is unclear what the "control" designation represents in the various results provided in figures figure 5-7.

8. The authors need to critically check all of the figures and figure legends to assure that all numbering is correct (See below).
9. The figure legends need to better described the data that is presented.

Minor points

1. In the figure legend for figure 1, the second mention of fig. 1c is not correct --it should be labeled Fig. 1e.
2. In the main text, figure 3e on the bottom of page 6 is not the correct annotation for what is being described. I think this should be figure 3d. In addition, figure 3e is not referenced in the manuscript.
3. In figure 4a-- the sequences for the shRNA targeting fibronectin are not provided. These were also not provided in the supplementary methods. It is also unclear what is in the lanes labeled "sh cont" in this figure.
4. In figure 6i, data is provided for CAL166. Why is only this portion of the data assessed for CAL166? There is no mention of this data and the manuscript.
5. In figure 7e-- what is shown in the lane 1 that is labeled as control?
6. I do not see changes in the expression of any of the interventions assessed in figure S4a. $\beta 6$ does not appear to be present in figure S4a.
7. The authors have to clarify exactly what is meant by the asterix use in the bar graphs in figures 3e, 6f, 7d, 7g. The figure legends indicate that these asterix represent confidence intervals estimates form linear fit, whereas the methods suggest that these represent p-values.

Reviewer #2 (Remarks to the Author): Expert in ECM and fibronectin

This paper reports the relative abundance of proteins in SDS-nonextractable extracellular matrix (ECM) of fibroblasts cultured from head and neck carcinomas as assessed by a non-intensity-based mass spectrometric method, identification of $\alpha 6 \beta 6$ and $\alpha 9 \beta 1$ as integrins responsible for carcinoma cell migration on such matrices, importance of cellular fibronectin for such migration, and a correlation of cellular fibronectin deposition in carcinoma tissue sections with aggressiveness of the tumors.

The mix of data is original and interesting, particularly regarding integrin usage.

The paper suffers from several shortcomings, however.

It is not clear how fibronectin deposition in tissue sections was scored. The numbers in Fig. 1A do not add up in any rational way. In 1A, scoring was apparently plus/minus. However, in Fig. 1C, scoring appears to be below and above median. The staining in Fig. 1B is only representative of a positive hotspot. 1B should be expanded to show the whole range of scoring. Fig. 1D shows lots of Western blots labeled by an unexplained system with no indication of what features are important and how those features were used.

The authors do not remark on what proteins are missing from the ECM. For instance, one might expect periostin (POSTN) to be present alongside its paralog, TGFBI (e.g., Kudo Y et al, Cancer Res, 2006). Was it really missing? The immunostaining of sections should be expanded to EDB+ fibronectin, TGFBI, and POSTN.

The migration data are convincing and quite beautiful.

Minor: TGFBI rather than TGFbetaI. Otherwise, the reader might confuse TGFbetaI with TGFbeta1.

Reviewer #3 (Remarks to the Author): Expert in integrin signalling

This paper from the Obbergen-Schilling lab presents data characterising the contribution made by the extracellular matrix (ECM) generated by stromal fibroblasts of head and neck cancers to the migratory behaviour of these cancer cells, and the integrins that involved in sensing these migrational cues.

The study is very interesting and presents findings that are of importance in regard of the role played by the deposition of proteins by carcinoma-associated fibroblasts (CAFs) and how these contribute to the invasive behaviour of cancer cells. The study makes a novel and important point about the contribution of the stromally generated ECM to cancer progression. I.e. that it is the oncofoetal fibronectin from carcinoma associated fibroblasts that is key to progression of head and neck squamous cell carcinoma.

Generally speaking the experimentation appears to be well-conducted and properly controlled, and the interpretation of the results is reasonable.

I have raised a few points which I have listed below, and I feel that if these are addressed the paper should be of interest to the readership of Nature Communications.

Major points:

1. I find the presentation of the cell migration data, and the description of how these have been obtained to be rather difficult to divine. Indeed from the data that are visible, I really do not understand how the authors can convince the readers that carcinoma cells move as clusters or 'collectives' on cell-derived matrices.

A) Firstly the authors should present representative movies to illustrate the type of migration displayed by carcinoma cells for a selection of key conditions;

B) The description of how the directionality ratio was obtained (page 8 of the supplement) and what this metric means is not clear to me at all;

C) Re the 'testing model' of Fig. 3d, it is not clear to me why or how this was performed;

D) I don't understand how the track-plots presented in Fig. 3c (for example) represent collective movement. To me there appears to be no concerted movement of the tracks to indicate collective movement. Also, what is denoted by the different grey levels of the lines in the track plots?

E) Why are there no stats for the velocity measurements. I think that the authors are putting store in the velocity as well as the directionality, measurements - so why no stats for the velocity whereas they are present for the directionality?

2. I gather from the text that the reason that the authors conducted a proteomic comparison of the ECM deposited by CAFs and TIFs was to justify the use of TIFs-derived ECM for their cell migration studies. However, from the data that are presented it may not be possible to determine how similar (or different) these preparations really are.

A) Is it possible to conclude using label-free proteomic approaches and only 2 biological replicates that FN1 is the most abundant ECM glycoprotein deposited by CAFs and TIFs?

B) the paper concludes that oncofetal (rather than plasma) FN is the major form of the protein in the ECM deposited by CAFs and TIFs, but can the mass spec approach discriminate between these isoforms? It ought to be able to, but these data don't appear to be presented.

3. One of the conclusions of the study is that the migratory behaviour of carcinoma cells is dependent on stromally deposited onco-FN and engagement of this with α v β 6 and α 9 β 1 integrins in the cancer cells, and that this is what drives cancer invasiveness and poor prognosis. The cell migration analysis that the authors have conducted is nice (given that they can resolve my points about presentation/analysis of the data), but this type of movement has not been linked to invasiveness in this study. It would be nice to see some experiments using organotypic approaches in which plugs of collagen have been preconditioned with CAF/TIFs (or onco-FN knockdown CAF/TIFs) to determine whether collective invasive behaviour requires stromal onco-FN expression.

Minor points:

1. The figure legends are a somewhat inadequate. For a Nature journal these could be expanded considerably. For instance: A) what are the colours in Fig. 1c? B) what is dFCS in Fig. 4a. I think it might be dialysed Foetal Calf Serum.. maybe not!
2. There is no loading control in Fig. 1d. How do the authors normalise for loading of these gels?
3. Erk1/2 is a strange choice for a loading control in Fig. 1e
4. I cannot see the staining for EDA-FN that is referred to in the text for Fig. 2b. The text says 'FN-EDA is incorporated in all the FN fibres' , but the figures present 'FN' staining. Also, where are the data indicating that 'EDA domains completely co-localised with staining of total FN'?
5. The initialism 'TDM' is not defined. Is this TIF-derived matrix? Maybe?
6. The text states that there is a 1.8-fold increase in $\beta 6$ transcripts(supplementary Fig. 4a). Should this be supplementary 4b?
- 7.. On page 5 line 7, I think that Fig 1c should be Fig. 1e.
8. Page 5 line 4 - I don't agree 'This pattern [necessarily] reflects the dynamic remodelling of the ECM in these tumours'.
9. This manuscript is submitted to a Nature journal so should it not use British spelling? Noah Webster's aberrations are apparent throughout this manuscript!!!

Reviewer #4 (Remarks to the Author): Expert in migration and cancer

Directional migration of head and neck carcinoma collectives on fibronectin-rich stromal matrix relies on integrins $\alpha v\beta 6$ and $\alpha 9\beta 1$.

This study explores the role of integrins $\alpha v\beta 6$ and $\alpha 9\beta 1$ in the collective migration of squamous cell carcinoma cell lines on fibroblast derived matrices. Fibronectin expression is linked to poor prognosis. Further, the authors show that integrin engagement with the fibronectin network enhances the directionality and velocity of SCC collective migration. They also characterise both the composition and organisation of ECM networks from three fibroblast lines. The authors show that $\alpha v\beta 6$ engagement activates latent TGF- β in SCCs, however this is not the mechanism through which enhanced migration is achieved.

Although experimentally this work is reasonably sound, the study does not offer a significant advance in our understanding of the interplay between the tumour and its ECM. FN has been linked to bad clinical outcomes previously (e.g. Ioachim et al., 2002). The observation that organised fibroblast derived ECM networks enhance the persistent migration of tumour cells has been reported many times before (Goetz et al., 2011; Sada et al., 2016; Yang, Mosher, Seo, Beebe, & Friedl, 2011). Similarly the role on $\alpha v\beta 6$ in SCC migration on fibronectin has also been demonstrated (Thomas, Lewis, et al., 2001; Thomas, Poomsawat, et al., 2001). Although not before implicated in SSC, $\alpha 9\beta 1$ has also been reported to regulate migration in colon adenocarcinoma (Gupta & Vlahakis, 2009) melanoma (Lydolph et al., 2009) and neutrophils (Shang, Yednock, & Issekutz, 1999). Beyond engagement of fibronectin, there is also little mechanistic insight into the role of these specific integrins. Overall, this study is too lightweight and not of sufficient novelty to be published in Nature Communications.

Additional specific comments

1. The authors conduct matrisome analysis of two HNSCC CAFs and a HTERT immortalised fibroblast. Can authors please indicate the disease state (if any) and site of origin of the later and expand on the reason for this comparison?
2. Western blot loading controls missing from figure 1D.
3. The authors show that putative invadosome structures are present at cell/ECM interface Figure S2A but do not provide a control such as uniform FN or gelatin coating. Does the fibrillar FN matrix promote invadopodia relative to control situations?
4. The mean squared displacement analysis and its use in describing persistent migration is not well-explained.
5. Can the authors please expand on their reason for interest in FN-EDB vs FN-EDA domains? Is

there anything known about the functional difference between the two, and any hypothesis around why FN-EDA would be incorporated into matrices but not FN-EDB? The authors should re-express different shRNA resistant FN isoforms to address their functional importance.

6. Is integrin $\alpha 9$ linked to disease outcome in HNSCC?

7. There is no proposed mechanism linking integrin $\alpha 9$ to migration or cell polarity: are key regulators such as Cdc42, Rac1, or RhoA regulated by integrin $\alpha 9$ in this context? Also, cofilin has been implicated in controlling persistent migration on FN via integrin $\alpha V\beta 3$ (Danen group 2005), is something similar happening downstream of $\alpha 9\beta 1$?

Reviewers' comments and authors' replies

We are sincerely grateful to the 4 reviewers with complementary expertise for agreeing to review this study and for highly insightful comments which have helped us to significantly improve the revised manuscript. Please find below our point-by-point reply in blue. Details regarding new results and modifications that have been made to the manuscript & figures are indicated in red.

Reviewer #1 (Remarks to the Author): Expert in HNSCC and clinical analysis

This is an interesting manuscript that assesses contributions of micro-environment in the directional movement of head, neck squamous cell carcinomas. They identify oncofetal fibronectin as a key component of the matrix and associate its overexpression with poor outcome in patients with head neck squamous cell carcinomas. The findings are novel and of interest to a broad audience. There are several questions about the data and methodology that need to be addressed. A key question is why transform fibroblasts were used, rather than cancer associated fibroblasts for all of the experiments in the manuscript. Details about the figures and controls are inadequate. The statistical approach is not well described either in the main manuscript or supplementary methods for several of the experiments. While the conclusions at the authors drive are congruent with the data presented, there is insufficient controls to make firm statements about precise mechanisms being investigated in this work. While this manuscript is potentially interesting, several issues need to be addressed.

Major issues include:

1. Figure 1 details results from immunostaining for fibronectin in a cohort of head neck cancer patients on a tissue microarray. The results show an association between fibronectin expression and TNM stage, tumor grade, and survival. A key issue with these findings is the fact that these experiments were performed on tissue microarrays. The construction of TMA's typically focuses on identifying areas within a tumor that are densely populated by cancer cells. It is unclear whether the results from the TMA reflect the expression of fibronectin in the remainder of the tumor. Although the supplementary data states that whole sections were compared for 40 cases, results from this comparison are not reported in this manuscript. In addition, while the supplementary data section includes details about the statistical analyses performed in these experiments, only disease-free survival is provided in the manuscript. Overall survival and local recurrence or not given. Moreover, while it is indicated that both unique variant and multivariate analyses were performed, results of multivariate analysis are not provided anywhere in the manuscript.

As pointed out by this reviewer, the association of FN expression with clinical parameters was based on results from analyses of tissue microarrays. The TMA configuration allows high-throughput profiling of protein expression in tumour specimens. In the case of stromal markers, we agree that it is important to bear in mind that TMA tumour biopsies are generally representative of areas rich in tumour cells. However, in our study the TMA construction by the pathologists took into account the chief objective of this study and therefore was mainly focused on the stromal component inside each tumour. Thus, stromal components were present in nearly

all of the histospots analyzed. In a sampling of 262 spots, 8.0% were scored as >50% carcinoma cells. To confirm the observed pattern of intratumoural FN expression we performed staining on a set of whole sections. Indeed, the FN expression observed in our TMA spots reflects expression of FN in the rest of the tumour tissue thus supporting our conclusions. Representative images of these stainings have been added to the Supplementary Figures (Fig. S1b and c) and mentioned in the results section of the revised manuscript (p. 5).

In addition to disease-free survival in the initial manuscript, we now provide the overall survival analysis and the results of the multivariate analysis (Fig. 1c and Supplementary Fig. S1a). Note that disease-free survival is defined as any disease recurrence (local, regional, or distant), but death is censored (not included).

2. In addition, the authors suggest that fibronectin is primarily expressed by stromal fibroblasts. Based on Western blot drive expression patterns in primary tumors, cancer cell lines, and demoralized human fibroblasts. While this may be the case, I'm not sure they can make this conclusion based on the data presented.

Indeed, FN is also expressed by other cells in the stroma of these tumours (i.e. endothelial cells, immune cells). Cellular FN expression by angiogenic endothelial cells has been largely documented elsewhere (Neri D and Bicknell R, Nature Reviews Cancer, 5: 436-446, 2005 and references therein). However, we focused on fibroblasts as they represent the major matrix-producing cells of the stroma. This has been clarified in the revised manuscript (p.5). In addition, CD31 and FN staining in nearby sections of the same tumour has been added to Supplementary Fig. S1c to illustrate blood vessel density and the modest contribution of endothelial cell-produced perivascular FN, and plasma FN to the cellular (EDA+) FN-rich stroma.

FN expression was low in tumour epithelial cells. This is consistent with published findings. For example, in an immunohistochemical study of integrins and their ligands in a smaller cohort of oral SCC (n=40), Fabricius et al. reported elevated expression of FN in the stroma of oral SCC, but none in the tumour cells (Fabricius, E.-M. et al. Exp. Ther. Med. 2: 9-19, 2011), now cited in the text (p.5).

3. In figure 2, the authors present data showing proteins deposited into matrix by transform fibroblasts and cancer associated fibroblasts. It would be of interest to know whether cancer associated fibroblasts also have patterns of fibronectin expression, similar to transform fibroblasts (as part of Western blot in figure 1E). If possible, it will be important to include a control of non-transformed fibroblasts to assure that the patterns of expression are specific to the cancer microenvironment.

Indeed, the cancer associated fibroblasts display a pattern of FN expression by western analysis that is similar to the telomerase-immortalized fibroblasts. Cultured normal fibroblasts secrete and assemble FN. This can be expected as serum-containing medium provides a "pathological" context that can trigger expression of oncofoetal proteins. However, the levels of FN produced by normal fibroblasts are considerably lower than those observed for TIFs or CAFs.

As requested by the reviewer we have added western analysis of medium and lysates of cancer associated fibroblasts and normal skin fibroblasts in Fig. 1e for comparison with TIFs (Fig. 1f and p.5).

4. Is it possible for the authors to assess whether fibronectin transcripts also contain more EDA domains, primary tumor samples.

It would be possible to determine the expression of EDA-containing transcripts in RNA from primary human tumours. However, we believe that the protein expression and distribution are more relevant for this study. Therefore, we determined expression of the protein in the stroma of human tumours using a monoclonal antibody specific for the EDA sequence (IST-9), as shown in Fig. 1b. EDA-containing FN is known to be highly expressed in tumour microvasculature (Villa, A et al. Int J Cancer, 122:2405-13, 2008). Immunofluorescence co-staining of EDB-FN and EDA-FN in CAFs has been added to revised Fig. 2b and mentioned on p.6.

5. In figure 3A, the difference between organization of CAL 133 on extracellular matrix does not look significantly different from those with no coat. A better image or multiple images should be provided to show this difference.

We have replaced the images of CAL33 and CAL166 cells on plastic and cell-derived matrix.

6. In figure 4, the requirement for fibronectin expression is assessed by knocking down fibronectin in transform fibroblasts and assessing effect on matrix formation. The authors state transform fibroblasts in which fibronectin is knockdown using shRNA do not produce a matrix. It would be important to know whether knocking down fibronectin in transform fibroblasts affects cellular activities essential to normal function and viability of the cells. If the cells are "sick" they may not produce a matrix solely because they lack fibronectin.

Since FN-knock down in TIFs has never been reported, we compared the morphology and growth rate of cells stably expressing Control or FN-targeting shRNA sequences. It can be seen from these results that the FN-deficient cells appear normal and FN silencing has only a minor effect on cell proliferation. We have mentioned this in the text and included these results in Supplementary Fig S5.

It is noteworthy that FN-null mouse embryo fibroblast lines have previously been generated in different laboratories (Sottile, J. et al, J Cell Sci., 1998; Sakai, T., et al. Nat. Med., 2001; our unpublished results) with no reported effect on cell viability. Moreover, we have not observed a compromising effect of FN depletion on the viability of FN-deficient endothelial cells (Cseh et al., J. Cell Sci., 2010) or glioblastoma cells (Serres, E. et al., Oncogene, 2015).

7. It is unclear what the "control" designation represents in the various results provided in figures 5-7.

The designation of "control" has been defined in the Legends to Figs. 5-7.

8. The authors need to critically check all of the figures and figure legends to assure that all numbering is correct (See below).

The figure numbering has been verified.

9. The figure legends need to better describe the data that is presented.

The Figure legends have been revised, as requested by reviewers.

Minor points

1. In the figure legend for figure 1, the second mention of fig. 1c is not correct --it should be labeled Fig. 1e.

This has been corrected.

2. In the main text, figure 3e on the bottom of page 6 is not the correct annotation for what is being described. I think this should be figure 3d. In addition, figure 3e is not referenced in the manuscript.

The description of Fig. 3d is now coherent and Fig. 3e has been referenced.

3. In figure 4a-- the sequences for the shRNA targeting fibronectin are not provided. These were also not provided in the supplementary methods. It is also unclear what is in the lanes labeled "sh cont" in this figure.

In the Methods section (**Lentiviral vector construction and transduction**) we indicated that the pLB2CPGm lentiviral vector harbouring FN- or luciferase-targeting (control) shRNA sequences have been described (Reference 42). In the revised version we have provided the sequences in the Supplementary Information section (Sequences of quantitative real-time PCR Primers and shRNA). The control shRNA (Sh cont) is now defined in the Legend to Fig. 4a.

4. In figure 6i, data is provided for CAL166. Why is only this portion of the data assessed for CAL166? There is no mention of this data and the manuscript.

CAL166 was used to validate our findings and not systematically included in the main Figures for space limitations. For homogeneity, we have removed the western blot analyses from Fig 6i and mentioned the results in the text.

5. In figure 7e-- what is shown in the lane 1 that is labeled as control?

The control corresponds to cells transduced with the empty vector (PLKO). This information has been added to the legend of Fig. 7e.

6. I do not see changes in the expression of any of the interventions assessed in figure S4a. $\beta 6$ does not appear to be present in figure S4a.

Indeed, these results show that there is no change of mRNA expression encoding the integrins represented on the array (revised Supplementary Fig. S7a). $\beta 6$ and $\alpha 9$ integrin subunits were not present on the "ECM and Adhesion molecules" RT2 profilerTM PCR array system from SABiosciences. Therefore we performed separate qPCR analyses to determine the effect of adhesion to TIF-derived ECM on mRNA expression encoding these integrin subunits in cells. Reference to these data, shown in revised Supplementary Fig. 74b, was mislabelled in the original manuscript and has been corrected in the revised text.

7. The authors have to clarify exactly what is meant by the asterix use in the bar graphs in figures 3e, 6f, 7d, 7g. The figure legends indicate that these asterix represent confidence intervals estimates form linear fit, whereas the methods suggest that these represent p-values.

Indeed, the information regarding statistical analyses merits further explanation. Modifications have been made in the Methods section (Statistical Analyses) of the manuscript (p. 17) and in the appropriate Figure legends. For clarity we consistently fixed the representation of asterisks for p-value and confidence interval. For both types of analyses, statistically significant data are indicated by *(p<0.05), ** (p<0.01), *** (p<0.001) or **** (p<0.0001).

Finally, regarding the question of why transformed fibroblasts were used, rather than cancer associated fibroblasts for our experiments, please refer to our response to Reviewer #3, point 2. Use of TIF-derived matrix yields more reproducible results for *in vitro* migration experiments.

Reviewer #2 (Remarks to the Author): Expert in ECM and fibronectin

This paper reports the relative abundance of proteins in SDS-nonextractable extracellular matrix (ECM) of fibroblasts cultured from head and neck carcinomas as assessed by a non-intensity-based mass spectrometric method, identification of α 5 β 1 and α 9 β 1 as integrins responsible for carcinoma cell migration on such matrices, importance of cellular fibronectin for such migration, and a correlation of cellular fibronectin deposition in carcinoma tissue sections with aggressiveness of the tumors.

The mix of data is original and interesting, particularly regarding integrin usage.

The paper suffers from several shortcomings, however.

It is not clear how fibronectin deposition in tissue sections was scored. The numbers in Fig. 1A do not add up in any rational way. In 1A, scoring was apparently plus/minus. However, in Fig. 1C, scoring appears to be below and above median. The staining in Fig. 1B is only representative of a positive hotspot. 1B should be expanded to show the whole range of scoring.

Regarding the numbers in Fig. 1a, there was an inversion in the number of moderately and poorly differentiated FN-positive tumours. The numbers should be 110 and 50, respectively. This has been corrected and the figures now add up. Thank you for pointing this out.

Indeed the scoring of tumours was plus/minus in Fig. 1b and Fig. 1c. We have added representative FN stainings of histospots in Figure 1b to show the range of scoring. Labels on the curves in Fig. 1c have been changed to low and high FN expression corresponding to (score 0-1) and (score 2-3), respectively, as defined in the revised legend.

Fig. 1D shows lots of Western blots labeled by an unexplained system with no indication of what features are important and how those features were used.

These data and their relevance have been presented more clearly in the text (p. 5) and legend to Fig. 1e of the revised manuscript.

The data in Fig. 1D correspond to western blot analyses of FN expression in lysates of tissue samples from 60 human HNSCC (described in Supplementary Information). Each blot includes 6-8 tumour samples (patient number indicated above blots) and a control lysate (CAL33 cells). As requested by Reviewers 3 and 4, actin staining was added to the Figure which provides a control of sample integrity and cellularity of the tumour samples.

Western analysis allows not only a relative quantification of FN levels in human tumour tissue but also provides a snapshot of the proteolysed state of the FN molecule (discussed in our response to comment 8 of Reviewer 3).

The authors do not remark on what proteins are missing from the ECM. For instance, one might expect periostin (POSTN) to be present alongside its paralog, TGFBI (e.g., Kudo Y et al, Cancer Res, 2006). Was it really missing? The immunostaining of sections should be expanded to EDB+ fibronectin, TGFBI, and POSTN.

General comment regarding proteins missing from the ECM:

Thank you for this interesting comment. Here we identified the matrix and matrix-associated molecules present in stromal fibroblasts (CAFs) from head and neck patient samples. Most of these molecules have previously been found in the matrix of lung and colon cancers (<http://matrisomeproject.mit.edu/>). However, some of the matrix components present in the matrix of colon and lung tumours were absent, or detected at very low levels, in the HNSCC-associated fibroblast matrisome. Some of the missing components correspond to different members of the same protein family that may display tissue-specific expression. We have compiled a list of major lung and colon tumour matrisome components missing from the HNSCC-associated fibroblast matrisome that is available upon request. To respect space limitations, and to stay focused, we have not discussed this point in the revised text.

Response to question regarding periostin and immunostainings:

First, we confirm (based on protein sequence alignment) that the peptides detected in our MS analyses correspond to TGFBI, and not periostin. TGFBI is one of the top 20 proteins identified (16, 11 and 10, respectively for CAF2, TIF and CAF1). The percent of sequence coverage for TGFBI is 69%, 57% and 75% for CAF2, TIF and CAF1, respectively.

Periostin was detected, but only in the CAF1 matrisome, at very low levels (position 348 of 376 proteins at 4% FDR and 18% coverage). In this case, we estimate using emPAI scores that TGFBI presents 276 more copies than POSTN. Therefore, we do not believe that there are problems with detection of the protein in our proteomic analyses. The low levels/lack of periostin in our fibroblast matrisomes could be explained by inter-/intra-tumoural heterogeneity (i.e. low expression by the CAFs isolated in our study) or by a requirement for interactions between the fibroblasts and tumour cells. Similar to findings in lung cancer, periostin was recently reported to be upregulated in head and neck cancer-associated fibroblasts in response to TGFβ3 (Qin X et al. Scientific Reports, February 2016). We would favour a combination of the latter 2 hypotheses.

Immunostaining of periostin in whole sections of HNSCC, as requested, revealed variable expression in the stromal compartment. The Vectastain ABC signal amplification Kit was used for detection of periostin staining. Although interesting for future investigations of tumour-stroma crosstalk, periostin is a presumed ligand for avb3, avb5 and a6b4 integrins which are not involved in the motility of HNSCC cells described in the present study.

We have added a statement regarding periostin in the discussion section (p.12), and representative images of periostin staining in HNSCC. In addition, we have included stainings of TGFBI and FN in nearby sections of the same human tumours for comparison (Supplementary Figs. S10 and S11).

Staining of EDB-FN has previously been shown to be abundantly expressed in the vasculature and in the stroma of various head and neck tumours (Birchler, MT et al, The Laryngoscope, 2003, Mhawech, P et al, Oral Oncol., 2004, Schwager, K. et al., Head & Neck Oncology, 2011) and absent in normal tissues. In these reports, stainings were performed on cryosections of frozen tumours. In our hands, staining of EDB-FN in formalin-fixed paraffin-embedded tissue sections using different lots of commercially available antibodies yields similar results. However, despite major efforts to optimize conditions, the background staining is elevated. Therefore, we have not added images to the revised manuscript. We do not believe that these results are essential to our study. However, as immunocytochemistry is possible with an EDB-specific antibody, we have added immunofluorescence co-staining of EDB-FN and total FN in the matrix of carcinoma associated fibroblasts (Fig. 2b).

Staining of TGFBI, a major glycoprotein component of our CAF matrisomes, was performed on sections of human tumours and representative images to accompany the text have been added to the Supplementary Figures. (p12, Supplementary Figs. S10,11).

The migration data are convincing and quite beautiful.

Minor: TGFBI rather than TGFbetaI. Otherwise, the reader might confuse TGFbetaI with TGFbeta1.

TGFβI has been replaced by TGFBI in the text.

Reviewer #3 (Remarks to the Author): Expert in integrin signalling

This paper from the Obbergen-Schilling lab presents data characterising the contribution made by the extracellular matrix (ECM) generated by stromal fibroblasts of head and neck cancers to the migratory behaviour of these cancer cells, and the integrins that involved in sensing these migrational cues.

The study is very interesting and presents findings that are of importance in regard of the role played by the deposition of proteins by carcinoma-associated fibroblasts (CAFs) and how these contribute to the invasive behaviour of cancer cells. The study makes a novel and important point about the contribution of the stromally generated ECM to cancer progression. I.e. that it is the oncofoetal fibronectin from carcinoma associated fibroblasts that is key to progression of head and neck squamous cell carcinoma.

Generally speaking the experimentation appears to be well-conducted and properly controlled, and the interpretation of the results is reasonable.

I have raised a few points which I have listed below, and I feel that if these are addressed the paper should be of interest to the readership of Nature Communications.

Major points:

1. I find the presentation of the cell migration data, and the description of how these have been obtained to be rather difficult to divine. Indeed from the data that are visible, I really do not understand how the authors can convince the readers that carcinoma cells move as clusters or 'collectives' on cell-derived matrices.

A) Firstly the authors should present representative movies to illustrate the type of migration displayed by carcinoma cells for a selection of key conditions;
Representative films have been uploaded.

B) The description of how the directionality ratio was obtained (page 8 of the supplement) and what this metric means is not clear to me at all;

The description of the analysis of cell migration has been modified in the revised Supplementary Information (p10).

Cell migration results from a combination of random (or Brownian) movement, and persistent movement in one direction. Thus, if a cell migrates more randomly, directionality decreases and vice versa. To determine the relative contribution of these two components (directionality ratio ρ_{Dir}), we used the experimental Mean Square Displacement (MSD) curves. Thus, we fitted the MSD curves with a linear combination of Brownian and constant speed movements. Given the fact that the Brownian movement is linear with time and the constant speed movement is quadratic with time, we defined the linear movement ratio of each cell as ρ_{Dir} ($MSD(t, \rho_{Dir}, D) = (1 - \rho_{Dir})4Dt + \rho_{Dir}v^2t^2$, where v is the mean instantaneous speed) such that it varies from 0 (i.e. Brownian movement) to 1 (directional movement) and is used as a directionality metric.

C) Re the 'testing model' of Fig. 3d, it is not clear to me why or how this was performed;

A testing model was used to confirm that our metric is robust and consistent. Thus, we simulated tracks as a mixture of Random walk and constant speed movement. More precisely the movement was built as:

$$P_k(x, y, t + \Delta t) = P_k(x, y, t) + V_m * \Delta t \left((1 - \widetilde{\rho_{Dir}}) * R_1 * \begin{bmatrix} \cos(R_2 * 2\pi) \\ \sin(R_2 * 2\pi) \end{bmatrix} + \widetilde{\rho_{Dir}} * \begin{bmatrix} \cos(\theta_k) \\ \sin(\theta_k) \end{bmatrix} \right)$$

R_i are randomly chosen at each iteration from 0 to 1 and θ_k a constant per track. By varying the directionality ratio in our simulations, we could link ρ_{Dir} obtained from the analysis to $\widetilde{\rho_{Dir}}$ defined as a parameter of the simulation. The sigmoidal curve was expected.

This has been described in Supplementary Experimental Procedures, Analysis of cell migration (p. 10)

D) I don't understand how the track-plots presented in Fig. 3c (for example) represent collective movement. To me there appears to be no concerted movement of the tracks to indicate collective movement.

Tracks do not display concerted movement since the individual tracks represent movement of non-dividing cells within different groups, as indicated in the Analysis of cell migration section on p9 of Supplementary Information.

Also, what is denoted by the different grey levels of the lines in the track plots?

The different grey levels (rather than colours) denote different tracks. This has been defined in the Figure legends.

E) Why are there no stats for the velocity measurements. I think that the authors are putting store in the velocity as well as the directionality, measurements - so why no stats for the velocity whereas they are present for the directionality?

The value obtained from the “QuanTrack” MATLAB script for velocity measurements represents the speed distribution of all cells monitored in a given analysis. This distribution is Poissonian-like, therefore by definition the standard deviation is equal to the mean value and error bars are irrelevant. This explanation has been added to the Supplementary Experimental Procedures, Analysis of cell migration (p10).

2. I gather from the text that the reason that the authors conducted a proteomic comparison of the ECM deposited by CAFs and TIFs was to justify the use of TIFs-derived ECM for their cell migration studies. However, from the data that are presented it may not be possible to determine how similar (or different) these preparations really are.

Indeed, the use of TIF-derived matrix yields more reproducible results for *in vitro* migration experiments. To address the concern of this reviewer, a comparative analysis of the global proteomes of the different ECM used in this study was performed. Pearson coefficients between the proteins identified by LC-MS are : 0.76 and 0.84 for CAF1 vs TIF and CAF2 vs TIF, respectively. These results have been included in the Supplementary Figures (Fig. S2e).

A) Is it possible to conclude using label-free proteomic approaches and only 2 biological replicates that FN1 is the most abundant ECM glycoprotein deposited by CAFs and TIFs?

Quantitative data in the table below clearly show that FN1 is among the most abundant protein detected in our analyses. This table can be added upon request, however we do not believe that this is necessary as the **Supplementary Table S1** provides the Mascot results and the molar % (emPAI score/ total emPAI score) for identified proteins at 1% FDR.

		Score	Position in the identified proteins	Number of Matches	emPAI score	%Mol
CAF1	FN Value	25598	3	1574	7	3,8
	Range	(21-42035)	(1-204)	(1-3344)	(256-0.01)	(10-3E-3)
	FN classification	3	3	3	11	5
CAF2	FN Value	28633	1	2757	13	1,1
	Range	(28633-21)	(1-239)	(2757-2)	(34-0.01)	(42-2E-3)
	FN Classification	1	1	1	5	11
TIF	FN Value	43148	1	2261	32	4,7
	Range	(43148-19)	(1-474)	(2261-1)	(32-0.01)	(9-3E-3)
	FN Classification	1	1	1	1	3

B) the paper concludes that oncofetal (rather than plasma) FN is the major form of the protein in the ECM deposited by CAFs and TIFs, but can the mass spec approach discriminate between

these isoforms? It ought to be able to, but these data don't appear to be presented.

The proteomic analysis allows us to confirm the presence of FN-**EDB**-/**EDA**+ (isoform 1) and FN-**EDB**+/**EDA**+ (isoform15) in the CAF1, CAF2 and TIF ECM proteomes by the identification of peptides specific to these 2 isoforms. We did not detect peptides specific to the **EDA**-sequence.

These proteomic results are consistent with the immunofluorescence co-staining of total FN and FN-EDA (Fig. 2b) in TIFs and CAFs, as well as the qPCR results shown in Fig. 2d. Finally, FN-EDA is clearly abundant in the stroma of human tumours as determined by specific staining of the isoform (Fig. 1b).

Results of these proteomic and immunofluorescence staining analyses have been added to the manuscript (p. 6 and Fig. 2b).

3. One of the conclusions of the study is that the migratory behaviour of carcinoma cells is dependent on stromally deposited onco-FN and engagement of this with avb6 and a9b1 integrins in the cancer cells, and that this is what drives cancer invasiveness and poor prognosis. The cell migration analysis that the authors have conducted is nice (given that they can resolve my points about presentation/analysis of the data), but this type of movement has not been linked to invasiveness in this study. It would be nice to see some experiments using organotypic approaches in which plugs of collagen have been preconditioned with CAF/TIFs (or onco-FN knockdown CAF/TIFs) to determine whether collective invasive behaviour requires stromal onco-FN expression.

A major aim of the present study was to identify specific interactions between HNSCC cells and their stromally-generated ECM environment that regulate motility. For this reason we chose to focus on fibroblast-derived matrices rather than mono-component 3-D collagen lattices in an organotypic or spheroid invasion configuration in this study. Although collagen gels display a fibrillar architecture, they lack the chemical and organisation complexity of the tumoural ECM. Moreover, FN fibrillogenesis precedes, and is required for, collagen assembly by fibroblasts.

This being said, it is noteworthy that the suggested organotypic type assays in which collagen is embedded with TIF/CAF requires total and sustained knock out of FN expression/assembly in fibroblasts for results to be conclusive. This is not possible for early passage CAFs, and shRNA-/siRNA-mediated knockdown of FN in TIFs is not complete (FN accumulates over the 21-day course of the experiment).

Minor points:

1. The figure legends are a somewhat inadequate. For a Nature journal these could be expanded considerably. For instance: A) what are the colours in Fig. 1c? B) what is dFCS in Fig. 4a. I think it might be dialysed Foetal Calf Serum.. maybe not!

We would like to thank you, and all the reviewers, for their detailed review of our manuscript and for spotting the mistakes that escaped our proofreading. In the revised version, **Figure legends have been extensively revised.**

The abbreviation dFCS that refers to FN-depleted FCS serum (defined only in the methods section) has now been defined in the legend to Fig. 4a.

2. There is no loading control in Fig. 1d. How do the authors normalise for loading of these gels? This point has also been raised by Reviewers 3 and 4. Thus, detection of actin in the fixed amount of tumour lysates has been added to Fig. 1d, providing a control of sample integrity and cellularity of the tumour samples.

3. Erk1/2 is a strange choice for a loading control in Fig. 1e
Total Erk1/2 is a robust loading control that is routinely used in our experiments.

4. I cannot see the staining for EDA-FN that is referred to in the text for Fig. 2b. The text says 'FN-EDA is incorporated in all the FN fibres', but the figures present 'FN' staining. Also, where are the data indicating that 'EDA domains completely co-localised with staining of total FN'?

We apologize, this was poorly explained in the Figure legend.

In Fig. 2b, staining of total FN in TIFs is shown in green and staining of EDA with isoform-specific antibodies is in red. Therefore, their co-localization appears yellow. In the revised manuscript we have replaced this Figure with the co-staining of total FN with isoform-specific antibodies to EDA or EDB in TIFs and in a carcinoma associated fibroblast preparation (CAF2). Note that all of the fibers are yellow in the FN/EDA co-stainings. This is consistent with the proteomic analyses in which we did not detect peptides specific to the EDA- sequence.

5. The initialism 'TDM' is not defined. Is this TIF-derived matrix? Maybe?

Yes, TIF-derived matrix was abbreviated as TDM. However, to avoid excess abbreviations we have removed "TDM" from the revised manuscript.

6. The text states that there is a 1.8-fold increase in $\beta 6$ transcripts(supplementary Fig. 4a). Should this be supplementary 4b?

Yes, this has been corrected.

7.. On page 5 line 7, I think that Fig 1c should be Fig. 1e.

Yes, this has been corrected.

8. Page 5 line 4 - I don't agree 'This pattern [necessarily] reflects the dynamic remodelling of the ECM in these tumours'.

Our statement was based on studies of ECM remodelling in general and in the regulation of FN turnover by extracellular proteases in particular. For example, the group of Jane Sottile has shown that MT1-MMP can regulate ECM FN remodelling by promoting extracellular cleavage of FN which results in the appearance of multiple FN fragments, without altering the global structure of FN fibrils (Shi, F. and Sottile J, J Cell Science, 2011). We clearly observe an increase in faster migrating FN fragments in the tumour-associated lysates, as compared to the soluble protein in cell conditioned medium. This is consistent with upregulation of proteinase expression by malignant and non-malignant stromal cells and the fact that FN can be cleaved by different families of proteases and several MMPs (Bonnans, C. et al. , Nature Reviews, Mol. Cell Biol., 2014). Indeed, glycosylation and other post-translational modifications of FN can also

affect electrophoretic mobility of the protein (enhance the smear). FN is highly glycosylated and carbohydrate groups may be larger in a tumour context.

The description of this Figure has been modified in the revised manuscript to take into account the comments of this reviewer and those of Reviewer 2.

9. This manuscript is submitted to a Nature journal so should it not use British spelling? Noah Webster's aberrations are apparent throughout this manuscript!!!

The American English spelling has been changed to British spelling.

Reviewer #4 (Remarks to the Author): Expert in migration and cancer

Directional migration of head and neck carcinoma collectives on fibronectin-rich stromal matrix relies on integrins $\alpha v\beta 6$ and $\alpha 9\beta 1$.

This study explores the role of integrins $\alpha v\beta 6$ and $\alpha 9\beta 1$ in the collective migration of squamous cell carcinoma cell lines on fibroblast derived matrices. Fibronectin expression is linked to poor prognosis. Further, the authors show that integrin engagement with the fibronectin network enhances the directionality and velocity of SCC collective migration. They also characterise both the composition and organisation of ECM networks from three fibroblast lines. The authors show that $\alpha v\beta 6$ engagement activates latent TGF- β in SCCs, however this is not the mechanism through which enhanced migration is achieved.

Although experimentally this work is reasonably sound, the study does not offer a significant advance in our understanding of the interplay between the tumour and its ECM. FN has been linked to bad clinical outcomes previously (e.g. Ioachim et al., 2002). The observation that organised fibroblast derived ECM networks enhance the persistent migration of tumour cells has been reported many times before (Goetz et al., 2011; Sada et al., 2016; Yang, Mosher, Seo, Beebe, & Friedl, 2011). Similarly the role on $\alpha v\beta 6$ in SCC migration on fibronectin has also been demonstrated (Thomas, Lewis, et al., 2001; Thomas, Poomsawat, et al., 2001). Although not before implicated in SSC, $\alpha 9\beta 1$ has also been reported to regulate migration in colon adenocarcinoma (Gupta & Vlahakis, 2009) melanoma (Lydolph et al., 2009) and neutrophils (Shang, Yednock, & Issekutz, 1999). Beyond engagement of fibronectin, there is also little mechanistic insight into the role of these specific integrins. Overall, this study is too lightweight and not of sufficient novelty to be published in Nature Communications.

Additional specific comments

1. The authors conduct matrisome analysis of two HNSCC CAFs and a HTERT immortalised fibroblast. Can authors please indicate the disease state (if any) and site of origin of the later and expand on the reason for this comparison?

The immortalized fibroblasts used in this study were isolated from normal human skin explants. This has now been specified in the text. The matrix assembled by these cells is similar in composition and organization to the CAF-derived matrix and provides a reproducible substrate (throughout several passages in culture) for *in vitro* migration, as indicated in our response to point 2 of Reviewer 3 and in the revised manuscript (Supplementary Fig. S5).

2. Western blot loading controls missing from figure 1D.

Actin staining has been added to the Western analysis of Fig. 1d (see also comments to Reviewers 2 and 3 regarding this Figure).

3. The authors show that putative invadosome structures are present at cell/ECM interface Figure S2A but do not provide a control such as uniform FN or gelatin coating. Does the fibrillar FN matrix promote invadopodia relative to control situations?

As control, we have added a condition in which tumour cells were plated on adsorbed FN (Supplementary Fig. S4).

On an immobilized FN coat, tumour cells efficiently removed the protein from the coverslip. However, organization of the actin cytoskeleton was different in cells on immobilized FN as compared to cells on the fibrillar ECM. Further, cortactin staining was more diffuse and localized in lamellipodial-like protrusions, rather than invadosomal structures. Moreover, no accumulation of tumour cell-derived extracellular vesicles was observed on adsorbed FN.

These results would suggest that the fibrillar matrix promotes invadopodia relative to the coated protein. However, the 2 conditions are quite different and difficult to compare. On one hand the FN molecule (plasma FN) is on a rigid surface, in a partially extended conformation. On the other hand, the FN (cellular FN) is organized in extensible fibers intertwined with other matrix components.

4. The mean squared displacement analysis and its use in describing persistent migration is not well-explained.

This description has been improved in the revised Supplementary Experimental Procedures, Analysis of cell migration, p. 10.

Cell migration results from a combination of random (or Brownian) movement, and persistent movement in one direction. Thus, if a cell migrates more randomly, directionality decreases and vice versa. To determine the relative contribution of these two components (directionality ratio ρ_{Dir}), we used the experimental Mean Square Displacement (MSD) curves. Thus, we fitted the MSD curves with a linear combination of Brownian and constant speed movements. Given the fact that the Brownian movement is linear with time and the constant speed movement is quadratic with time, we defined the linear movement ratio of each cell as ρ_{Dir} ($MSD(t, \rho_{Dir}, D) = (1 - \rho_{Dir})4Dt + \rho_{Dir}v^2t^2$, where v is the mean instantaneous speed) such that it varies from 0 (i.e. Brownian movement) to 1 (directional movement) and is used as a directionality metric.

5. Can the authors please expand on their reason for interest in FN-EDB vs FN-EDA domains? Is there anything known about the functional difference between the two, and any hypothesis around why FN-EDA would be incorporated into matrices but not FN-EDB? The authors should re-express different shRNA resistant FN isoforms to address their functional importance.

We agree that this issue was not clearly presented in the submitted manuscript, thank you for pointing this out.

First, we were interested in knowing to what extent the observed staining in human tumours corresponds to cellular FN versus plasma FN, which circulates at high concentrations in the blood and could accumulate in tumour tissue through leakage from the bloodstream. Our results show

that the fibrillar FN in the stroma of human tumours contains the EDA domain and thus corresponds to cellular FN. Second, we were interested in determining the expression of FN-EDB and FN-EDA by CAFs/TIFs the matrix since the inclusion of EDA has been shown to expand the repertoire of integrins that can bind to FN, to include $\alpha 9\beta 1$. This has been highlighted in the results section on p. 10-11. No receptor has thus far been reported for the EDB, although its presence has been postulated to alter the conformation of the cell binding domain of FN (Ventura, Sassi et al. 2010).

Regarding their known functions, deletion of both EDB-FN and EDA-FN is embryonic lethal in mice (despite the maintenance of plasma FN). However, single Extra Domain-targeted deletion studies in the mouse have revealed roles for EDA in the morphogenesis of lymphatic valves, atherosclerosis and wound healing/fibrosis (reviewed in (White, Baralle et al. 2008)). In cancer, EDA-FN is a principal component of the premetastatic niche involved in recruiting bone marrow-derived progenitors.

Regarding the incorporation of the cellular FN isoforms in the matrix, both FN-EDB and FN-EDA are present in the CAF/TIF ECM. Proteomic analyses allowed us to confirm the presence of FN-EDB-/EDA+ and FN-EDB+/EDA+ in the CAF1, CAF2 and TIF ECM proteomes by the identification of peptides specific to these 2 isoforms. Peptides specific to the FN-EDA-sequence (i.e. plasma FN) were not detected. This has been added to the results (p. 6). Moreover, we have included isoform-specific immunofluorescence staining of EDB and EDA in CAFs and TIFs (revised Fig. 2b). The EDA domain completely co-localizes with staining of total FN, indicating that FN-EDA is incorporated in all of the FN fibers, whereas an FN-EDB-specific antibody recognized only a subset of fibers (Fig. 2b). FN transcripts expressed by TIFs contained relatively more of the EDA domain than the EDB domain (Fig. 2d).

We thank the reviewer for his/her comment but we sincerely believe that that the suggested experiments are not necessary for the general conclusions of the present study.

6. Is integrin $\alpha 9$ linked to disease outcome in HNSCC?

Information regarding $\alpha 9$ and clinicopathological correlations in HNSCC is relatively limited. Two studies on HNSCC patients in which ITGA9 expression data are publically available respond to this criterion:

- Kuriakose MA et al., Cell Mol Life Sci. 2004 Jun;61(11):1372-83.
- Chung CH et al., Cancer Res. 2006 Aug 15;66(16):8210-8.

Data from these publications (22 patients for Kuriakose MA et al. and 36 patients for Chung CH et al.) were extracted from the GEO DataSets. Our exploratory analyses revealed that ITGA9 may be linked to tumour grade ($p=0.033$), pT ($p=0.019$) and to HPV status ($p=0.015$). In contrast, no statistically significant correlation was found between ITGA9 expression and overall survival of patients, or between expression of the integrin subunit and recurrent disease. However, these conclusions are drawn from investigations of a very limited set of data and must be validated by further analyses on a larger set of patients.

These preliminary observations have been mentioned in the discussion section (p. 14).

7. There is no proposed mechanism linking integrin $\alpha 9$ to migration or cell polarity: are key regulators such as Cdc42, Rac1, or RhoA regulated by integrin $\alpha 9$ in this context? Also, cofilin

has been implicated in controlling persistent migration on FN via integrin α V β 3 (Danen group 2005), is something similar happening downstream of α 9 β 1?

We have not determined which integrin effectors operating downstream of α 9 β 1 integrin are involved in orchestrating the migration response in our system. This is an unmet and difficult challenge currently under investigation. However, as pointed out in our response to specific comment 3, the present experimental system involving a complex, compliant, fibrillar matrix rich in cellular FN is quite different from the classical systems used to study directional migration of individual cells (usually fibroblasts) on a rigid surface coated with immobilized plasma FN.

REVIEWERS' COMMENTS:

Reviewer #2 (Remarks to the Author):

This revised version corrects the numerous minor inconsistencies in the original submission, provides a detailed overview of molecules contributing to migration of HNCCs, and makes strong arguments for the unexpectedly large contributions of α v β 6 and α 9 β 1 integrins to directional migration.

Reviewer #3 (Remarks to the Author):

In general the revisions made by the authors have improved the paper. However, I have one remaining issue which could be dealt with by modifying the text and by toning-down claims. I don't think that the way that the mass spec experiments have been performed allow conclusions to be drawn re the relative levels of proteins in the CAF versus TIF samples, and it's important that the authors do not claim or imply this.

a) Mascot measures the quality of identification of a given protein, not the quality of quantification. The authors' response implies some confusion here.

b) There is only 1 replicate for the TIF matrisome.

c) One protein, protein (Titin, which I presume is quite abundant) has not been quantified in the TIF matrisome, why? See the 1st datasheet, which is called "TIF Mascot result". Very last line. If it were, would the conclusions change?

d) The matrisomes of CAF1 and CAF2 have very different amount of contaminants (ACTB, ACTG1...) and COL1A1, showing that reproducibility of the sample prep is not great. See fourth datasheet, which is called "Combined dataset" (also the datasheet where you can find FN1). I calculated the overall correlation (Pearson), which is 0.4, so not too high.

Save for this issue, the paper is very interesting and is suitable for the readership of Nature Comms.

Reviewer #4 (Remarks to the Author):

This work on the role of aligned fibronectin matrices on SCC invasion is improved from the original submission. The minor comments have all been reasonably addressed, however I still think that this work does not have level of conceptual advance or novelty that I would expect for Nature Communications. For this reason, I cannot recommend publication in Nature Communications and would suggest a more specialist journal.

Specific issues

1. Novelty/Conceptual advance: EDA fibronectin has been reported to be increased in many cancer types, including SCC (Heuser et al Oncogene 2016, Kelch et al JID 2015, Ou et al Carcinogenesis et al 2014, Sun et al Carcinogenesis 2014 and other studies including Kosmehl et al 1999 studying EDB fibronectin in SCC). More broadly, links between fibronectin expression and cancer phenotypes have been extensively documented. The data presented here is more extensive and has clearer links to outcome, but it is not transformative. The link between CAF generate matrix and persistent migration is also not new (Goetz et al., 2011; Sada et al., 2016; Yang, Mosher, Seo, Beebe, & Friedl, 2011). Finally, as stated in the original review, integrins α v β 6 and α 9 β 1 are well established fibronectin receptors and have been previously linked to cancer invasion.

2. Mechanism linking integrin α 9 β 1 to polarized cell migration: In their rebuttal the authors argue that pursuing this is beyond the scope of the current work and therefore do not address it. Given the marginal increase in our knowledge provided by the rest of the study, it is unsatisfactory that the authors have declined to try to find some new mechanistic insight. The experiments are not so

complex to perform.

Reviewer #5 (Remarks to the Author):

The authors have addressed the main concerns raised by the reviewers and is suitable for publication in Nature Communications.

The only remaining concerns is that the numbers in the table included in Figure 1A do not add up. It would be important to correct this information and make sure that the statistical analyses have been done with the proper numbers.

We sincerely thank the reviewers for their thoughtful evaluation of our study. Please find in red below our response to the questions and remarks concerning the revised version of our manuscript.

REVIEWERS' COMMENTS:

Reviewer #2 (Remarks to the Author):

This revised version corrects the numerous minor inconsistencies in the original submission, provides a detailed overview of molecules contributing to migration of HNCCs, and makes strong arguments for the unexpectedly large contributions of alphaVbeta6 and alpha9beta1 integrins to directional migration.

Reviewer #3 (Remarks to the Author):

In general the revisions made by the authors have improved the paper. However, I have one remaining issue which could be dealt with by modifying the text and by toning-down claims.

I don't think that the way that the mass spec experiments have been performed allow conclusions to be drawn re the relative levels of proteins in the CAF versus TIF samples, and it's important that the authors do not claim or imply this.

a) Mascot measures the quality of identification of a given protein, not the quality of quantification. The authors' response implies some confusion here.

We agree with the reviewer that the quantitation method used in our study, the exponentially modified protein abundance index (emPAI), is not the best available method for quantitative proteomics. However emPAI offers an approximate label free relative quantitation of the proteins in a mixture based on the number of observed and observable peptides. This emPAI index is calculated automatically by Mascot based on the publication by Ishihama and colleagues (Ishihama, Y., et al., Exponentially modified protein abundance index (emPAI) for estimation of absolute protein amount in proteomics by the number of sequenced peptides per protein, *Molecular & Cellular Proteomics* 4 1265-1272 (2005)). More information about emPAI can be found in the help of Mascot (http://www.matrixscience.com/help/quant_empai_help.html). Indeed, Mascot was originally developed to measure the quality of identification as pointed out by the reviewer, but recent versions of the software include also by default some label-free quantification as the emPAI to evaluate the protein abundance.

In this study we were unable to perform more precise label-free quantification of samples since less than 3 replicates were analysed per condition and samples were not all analysed at the same time. However, emPAI as we indicated in the Methods section, was used to estimate the protein abundance. The emPAI score is here the best label-free relative quantitation that can be used to approximate the protein abundance in our samples.

To avoid confusion for the readers we propose to eliminate the word "quantification" from the title of the Methods paragraph page 21 and replace the title by: "**Protein identification and estimation of abundance**". The estimation of protein abundance is quite confident for the most abundant proteins. In our opinion, we clearly show that FN1 is among the most abundant protein detected in our analyses with 4, 1 and 5 % molar respectively for CAF1, CAF2 and TIF.

b) There is only 1 replicate for the TIF matrisome.

For comparison with the CAF matrisomes 1 replicate of the TIF matrisome was used. This is one reason for using the emPAI index to approximate the relative abundance of identified proteins rather than a more accurate method, as mentioned above. We performed additional analyses of the TIF matrisome in an independent set of experiments.

c) One protein, protein (Titin, which I presume is quite abundant) has not been quantified in the TIF matrisome, why? See the 1st datasheet, which is called "TIF Mascot result". Very last line. If it were, would the conclusions change?

In our experiments we identified the TITIN in the TIF experiment with only 4 significant sequences among 554 putative sequences. In addition the emPAI index generated automatically by Mascot is missing, suggesting that the protein was a very weak hit. As the emPAI is calculated $10^{(\text{number of observed peptides}/\text{number of observable peptides})-1}$, the emPAI for TITIN is largely under 0.01. In summary, our identification of TITIN is probably false identification generated by the 1% false discovery rate and we don't believe that this changes our conclusions.

Having said that, it is not clear why the referee presumed that the TITIN protein was abundant in our preparations. The TITIN protein is not a known matrisome protein (<http://web.mit.edu/hyneslab/matrisome/>). Indeed, it is generally falsely identified by mass spectrometry and Mascot software because TITIN is such a large protein (around 4 million Daltons generating more than 4000 potential peptides, and the calculated emPAI should be less than 0,002) and may pick up several random matches in MS-MS spectrums. The 'titin effect', is very well known by proteomists where the very largest proteins are promoted because they randomly match a considerable number of peptides.

d) The matrisomes of CAF1 and CAF2 have very different amount of contaminants (ACTB, ACTG1...) and COL1A1, showing that reproducibility of the sample prep is not great. See fourth datasheet, which is called "Combined dataset" (also the datasheet where you can find FN1). I calculated the overall correlation (Pearson), which is 0.4, so not too high.

We don't think that the correlation between experiments can be directly calculated using the %mol from the data sheet, which is called « Combined dataset ». It is necessary to at least transform the value using for example a log2 to better evaluate the correlation. This transformation allows a normal Gaussian shape distribution and will make the distance from 1 to 0.5 (linear values) the same as 1 to 2. When correlation is calculated using log2(%mol) we found values between 0.68 (CAF1 vs. TIF) and 0.77 (CAF2 vs TIF; CAF2 vs. CAF1). We can visualize this correlation using the attached clustering (heat map.png) and scatter plots (multiscatterplot.png showing the Pearson correlation). Another way to evaluate the correlation between experiments is to use directly the peptide or protein intensity that can be calculated from each of the raw mass spectrometry files (data not shown in the paper). In this case, the Pearson correlations are 0.76, 0.80 and 0.84 for respectively CAF1 vs. TIF, CAF1 vs. CAF2 and CAF2 vs. TIF.

Save for this issue, the paper is very interesting and is suitable for the readership of Nature Comms.

Reviewer #4 (Remarks to the Author):

This work on the role of aligned fibronectin matrices on SCC invasion is improved from the original submission. The minor comments have all been reasonably addressed, however I still think that this work does not have level of conceptual advance or novelty that I would expect for Nature Communications. For this reason, I cannot recommend publication in Nature Communications and would suggest a more specialist journal.

Specific issues

1. Novelty/Conceptual advance: EDA fibronectin has been reported to be increased in many cancer types, including SCC (Heuser et al Oncogene 2016, Kelch et al JID 2015, Ou et al Carcinogenesis et al 2014, Sun et al Carcinogenesis 2014 and other studies including Kosmehl et al 1999 studying EDB fibronectin in SCC). More broadly, links between fibronectin expression and cancer phenotypes have been extensively documented. The data presented here is more extensive and has clearer links to outcome, but it is not transformative. The link between CAF generate matrix and persistent migration is also not new (Goetz et al., 2011; Sada et al., 2016; Yang, Mosher, Seo, Beebe, & Friedl, 2011). Finally, as stated in the original review, integrins $\alpha\beta6$ and $\alpha9\beta1$ are well established fibronectin receptors and have been previously linked to cancer invasion.
2. Mechanism linking integrin $\alpha9\beta1$ to polarized cell migration: In their rebuttal the authors argue that pursuing this is beyond the scope of the current work and therefore do not address it. Given the marginal increase in our knowledge provided by the rest of the study, it is unsatisfactory that the authors have declined to try to find some new mechanistic insight. The experiments are not so complex to perform.

First we would like to thank the reviewer for his/her comments on the manuscript. Indeed, increased expression of FN has previously been reported to be associated with various cancers and outcomes, as cited in the manuscript. However, we would like to underscore the fact that our manuscript provides novel biochemical and mechanistic information about the specialized ECM composition and architecture produced by tumor associated fibroblasts. We show that cellular FN isoforms are key and obligate components of this matrix and that this FN-rich fibrillar ECM (as opposed to plasma FN-coated rigid surfaces) provides specific ligand/integrin mediated interactions with tumor cells that promote their migratory activity. This information is conceptually important and to our knowledge has not been previously described at this level of detail. Indeed, the FN-binding integrins $\alpha v\beta 6$ and $\alpha 9\beta 1$ have been linked to cancer cell migration and invasion (as has $\alpha 5\beta 1$). However, previous in vitro studies have not taken the unique features and composition of the stromal ECM described here into account.

Importantly, while $\alpha v\beta 6$ has been described as upregulated in epithelial cancers (including head and neck squamous cell carcinomas) at the tumor/stromal interface, details regarding the mechanisms how the ECM regulates the migratory activity of the tumor epithelial cells via $\alpha v\beta 6$ has not been clear (Marsh, D. *et al.* J Pathol, 2011; Moutasim, KA *et al.*, J Pathol, 2011; Moutasim, KA *et al.* J Pathol, 2014). The present study is the first to highlight that this can be independent from $\alpha v\beta 6$ -mediated TGF β activation. Understanding that $\alpha v\beta 6$ effects on migration can be independent of TGF β activation differentiates it from other therapeutic approaches that have focused on inhibiting TGF β activity. This is important to understanding the potential for $\alpha v\beta 6$ -targeted therapeutics, especially important since an $\alpha v\beta 6$ mAb is in clinical development in fibrosis (<https://clinicaltrials.gov/ct2/results?term=bg00011&Search=Search>) and may have applications in oncology.

In conclusion, we hope that we have convinced the reviewer that our study is not a mere description of FN expression and its correlation with tumor outcomes and motility-promoting effects without providing novel information or raising any new issues that have not come up previously.

Reviewer #5 (Remarks to the Author):

The authors have addressed the main concerns raised by the reviewers and is suitable for publication in Nature Communications.

The only remaining concerns is that the numbers in the table included in Figure 1A do not add up. It would be important to correct this information and make sure that the statistical analyses have been done with the proper numbers.

We would like to thank the reviewer you for pointing this out. We regret this error and are not sure how it occurred. The numbers have been corrected in the present version of the manuscript and the analyses have been performed with the correct numbers. Importantly, the corrections do not change the p values.

To comply with the instructions in the Nature Communications Manuscript Checklist, indicating that Figures should not contain tables, we have removed these results from Figure 1 and included them as an independent “**Table 1**” using the Table menu in Word (with the ‘track changes’ feature) in the revised version of the manuscript (after Figure legends).